



# A coupled stochastic rainfall-evapotranspiration model for hydrological impact analysis

Minh Tu Pham[*1], Hilde Vernieuwe[2], Bernard De Baets[2], and Niko E. C. Verhoest[1]

[1]Laboratory of Hydrology and Water Management, Ghent University, Coupure links 653, 9000 Ghent, Belgium
[2]KERMIT, Department of Mathematical Modelling, Statistics and Bioinformatics, Ghent University, Coupure links 653, 9000 Ghent, Belgium

**Abstract**

A hydrological impact analysis concerns the study of the consequences of certain scenarios on one or more variables or fluxes in the hydrological cycle. In such exercise, discharge is often considered, as especially extreme high discharges often cause damage due to the coinciding floods. Investigating extreme discharges generally requires long time series of precipitation and evapotranspiration that are used to force a rainfall-runoff model. However, such kind of data may not be available and one should resort to stochastically-generated time series, even though the impact of using such data on the overall discharge, and especially on the extreme discharge events is not well studied. In this paper, stochastically-generated rainfall and coinciding evapotranspiration time series are used to force a simple conceptual hydrological model. The results obtained are comparable to the modelled discharge using observed forcing data. Yet, uncertainties in the modelled discharge increase with an increasing number of stochastically-generated time series used. Notwithstanding this finding, it can be concluded that using a coupled stochastic rainfall-evapotranspiration model has a large potential for hydrological impact analysis.

## 1 Introduction

Precipitation is the most important variable in the terrestrial hydrological cycle that determines soil moisture and discharge from a watershed. As such, it also impacts water management where generally the occurrences of extreme events, e.g. storms or droughts, which have very low frequencies, are of concern. Very long time series of precipitation are hence needed. Because this kind of data is not always available, one may consider using a stochastically-generated rainfall time series (Boughton and Droop, 2003). Stochastic rainfall models can be used to produce very long time series or to compensate for missing data from finite historical records (Wilks and Wilby, 1999). Several types of rainfall models have been proposed in literature. Onof et al. (2000) grouped all continuous rainfall models into four types: (1) meteorological models; (2) stochastic multi-scale models; (3) statistical models and (4) stochastic process models. Meteorological models are capable to describe the physical processes of all weather variables, including rainfall, by making use of very large and complex sets of equations. Numerical Weather Prediction and General Circulation Models are two common examples of this type of models. Stochastic multi-scale models describe the spatial evolution of the rainfall process regardless of scale factors. In general, these models involve an assumption of temporal invariance of rainfall over a range of scales (Bernardara et al., 2007). Statistical models, which can be used for simulating the precipitation trends, usually treat

---

*MinhTu.Pham@UGent.be





the occurrence and the amount of precipitation separately (Wilks and Wilby, 1999). The rainfall occurrence is represented by a sequence of dry and wet periods, usually simulated by Markov chains or Alternating Renewal Models. The precipitation amounts can be arbitrarily generated by making use of some popular distributions, e.g. the exponential (Todorovic and Woolhiser, 1975), the Gamma (Stern and Coe, 1984; Viglione et al., 2012) or the mixed exponential distribution (Woolhiser and Roldán, 1982; Wilks, 1998; Mason, 2004). Stochastic process models use simple assumptions of physical processes to simulate the hierarchical structure of the rainfall process. In this approach, only a limited number of parameters is needed (Verhoest et al., 2010). The Bartlett-Lewis (BL) (Rodriguez-Iturbe et al., 1987a) and the Neyman-Scott (Kavvas and Delleur, 1981) models are the most commonly used models of this type. In this study, we only focus on the BL models. These models have been applied successfully in different areas, such as Great Britain (Onof and Wheater, 1993; Onof et al., 1994; Cameron et al., 2000), Ireland (Khaliq and Cunnane, 1996), Belgium (Verhoest et al., 1997; Vandenberghe et al., 2010; Vanhaute et al., 2012), the United States of America (Rodriguez-Iturbe et al., 1987b; Velghe et al., 1994), New Zealand (Cowpertwait et al., 2007), Australia (Gyasi-Agyei, 1999; Heneker et al., 2001) and South-Africa (Smithers et al., 2002). The BL models are chosen in this study for three main reasons: (1) they show a good performance in all recent studies; (2) they are capable of generating time series at a sufficient fine time scale (less than 1 hour); (3) their calibration is easy given the limited number of parameters; and (4) they mimic well the stochastical behavior of the historical time series at Uccle (Verhoest et al., 1997; Vanhaute et al., 2012), which is used in this study.

Besides precipitation, the water balance is also highly influenced by the amount of water that is lost due to evapotranspiration. An accurate estimation of evapotranspiration is very essential for hydrological and agricultural designs, irrigation plans and for water distribution management (Droogers and Allen, 2002). The daily reference evapotranspiration is often modelled based on the Penman, Priestley–Taylor or Hargraeves equations; however, one major limitation of these models is that they require extensive input data, such as daily mean temperature, wind speed, relative humidity and solar radiation, which are not always available. Therefore, one may consider to rely on another approach based on stochastically-generated time series. More importantly, in order to obtain a correct evaluation of the water balance of a catchment and its discharge, these stochastic evapotranspiration data need to be consistent with the accompanying precipitation time series data (Pham et al., 2016). In this case, we can make use of the copula-based approach introduced in the work of Pham et al. (2016) in which the statistical dependence between evapotranspiration, precipitation and temperature is described by three- and four-dimensional vine copulas.

Many modelling approaches exist for simulating catchment discharge. The simplest models are the conceptual models in which several (non-)linear reservoirs are put in series and/or parallel. Well-known examples of such conceptual models are: the Hydrologiska Byräns Vattenbalansavdelning model (Bergström, 1995), the NedborAfstromnings Model (Nielsen and Hansen, 1973) and the Probability Distributed Model (PDM) (Moore, 2007). Alternatively, physically-based models are based on scientific knowledge of different hydrological processes and their interactions. Generally, these models contain many more parameters than the conceptual ones and require more input data, such as soil type, vegetation-related information, etc. Well-known examples of such models are the Soil and Water Assessment Tool (Arnold et al., 1998), the Système Hydrologique Européen (Abbott et al., 1986) and the Common Land Model (Dai et al., 2003). In this study, we do not intend to seek for the best hydrological model to assess our objective, but we opt for a model that is used in operational water management. More specifically, we will use PDM, as this model is used by the Flemish Environmental Agency (Cabus, 2008), and apply it to a catchment in Flanders, Belgium. The objective of this research is to assess whether the BL stochastically-generated rainfall and consistent evapotranspiration time series can be used for hydrological impact analyses. More specifically, we will evaluate different ways to apply stochastically modelled time series as forcing data to simulate the catchment's discharge. By increasing the number of stochastically-generated inputs to the model, we will assess the increase of uncertainty in modelled extremes and what portion of this increase can be attributed to the different stochastic generators. Section 3 first briefly introduces the coupled stochastic rainfall-evapotranspiration model and all the considered situations to simulate discharge from stochastic forcing data. Section 2 describes the





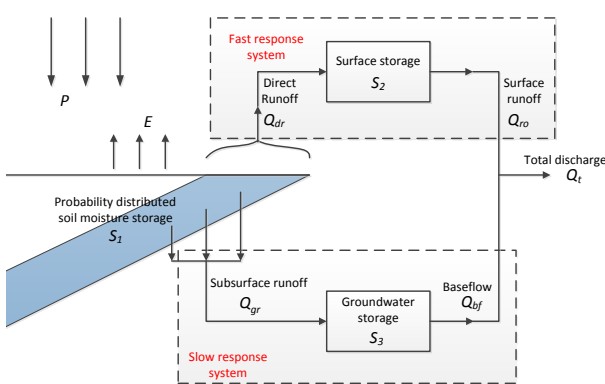

Figure 1: General model structure of the PDM (adapted from Moore, 2007).

historical records and all models used within this study. The discharge simulations from different scenarios are then evaluated in Section 4 allowing for assessing the impact of stochastic data on the simulation of discharge. Finally, conclusions and recommendations are given in Section 5.

## 2  Data and models

### 2.1  Historical data

This study uses observed time series measured in the climatological park of the Royal Meteorological Institute (RMI) at Uccle, near Brussels, Belgium. The data include time series of observed precipitation [mm] from 1898–2002, and mean daily temperature $T$ [°C] and daily reference evapotranspiration $E$ [mm/day] from 1931–2002. The time series of $E$ is derived using the Penman-Monteith equation. The precipitation data have been recorded with a time resolution of 10 min from 01/01/1898 to 31/12/2002 measured by a Hellmann–Fuess pluviograph (Démarée, 2003). This data set is quite unique in hydrology due to its extraordinary length with a sampling frequency of 10 minutes. Its quality is ensured consistently at a high level by using the same method of processing and measuring at the same location since 1898 (Ntegeka and Willems, 2008). This time series has been used in several studies (Verhoest et al., 1997; Vaes and Berlamont, 2000; De Jongh et al., 2006; Ntegeka and Willems, 2008; Vandenberghe et al., 2010; Vanhaute et al., 2012; Pham et al., 2013; Willems, 2013; Pham et al., 2016) and is used to calibrate the rainfall model as explained in Section 2.4. This time series has also been reprocessed to daily total precipitation [mm/day], further referenced to as $P$, for the period of 1931–2002, which is then used together with the time series of $T$ and $E$ for the construction of different stochastic models.

### 2.2  Probability Distributed Model (PDM)

PDM is a lumped rainfall-runoff model which basically conceptualizes the absorption capacity of soil in the catchment as a collection of three different storages (Moore, 2007; Cabus, 2008) (see Fig. (1): i.e. (1) a probability distributed soil moisture storage ($S_1$) based on a Pareto distribution of soil moisture capacity to separate direct runoff $Q_{dr}$ and subsurface runoff $Q_{gr}$; (2) a surface storage ($S_2$) to transform direct runoff into surface runoff; and (3) a groundwater storage ($S_3$) to convert subsurface runoff to baseflow. The input for $S_1$ is the net precipitation $(P - E)$, in which $P$ and $E$ are the precipitation and evapotranspiration, respectively. Further water loss from $S_1$ may be due to $Q_{dr}$ or $Q_{gr}$. The former is then converted to surface runoff $Q_{ro}$ through surface storage $S_2$, a fast response system involving a sequence of two linear reservoirs with small storage time constants $k_1$ and $k_2$. The direct runoff flow only happens when $S_1$ is completely filled. The





recharge to the groundwater, controlled by the drainage time constant $k_g$, is transfered into base-
flow $Q_{bf}$ through groundwater storage $S_3$, a slow non-linear response system with a large storage
time constant $k_b$. The sum of $Q_{ro}$ and $Q_{bf}$ equals the total discharge $Q_t$; note that a constant
flow which presents any returns or abstractions to or from the catchment, represented by a pa-
rameter $q_{const}$, also can be added. For a more detailed theoretical explanation and mathematical
description of the model, we refer to Moore (2007).

In this study, PDM is calibrated for the Grote Nete catchment using the Particle Swarm
Optimization algorithm (PSO) (Kennedy and Eberhart, 1995). This catchment, covering about
385 km² in the North of Belgium, has a maritime, temperate climate with an average precipi-
tation of about 800 mm/year (Vrebos et al., 2014). A time series of more than 6 years (from
13/8/2002–31/12/2008) at hourly time-step (precipitation, evapotranspiration and discharge) for
the catchment is available, in which the observations recorded during the period of 13/8/2002–
31/12/2006 are used for model calibration, while the remaining data (from 1/1/2007–31/12/2008)
are used for model validation.

## 144 2.3 Copula-based stochastic simulation of evapotranspiration and tem-
145 perature

### 146 2.3.1 Vine copulas

A copula is a multivariate function that describes the dependence structure between random
variables, independently of their marginal distributions (Sklar, 1959). The theorem of Sklar (Sklar,
1959) states that if $F_{12}(x_1, x_2)$ is the joint distribution function of two random variables $X_1$ and
$X_2$ with marginal cumulative distributions $F_1$ and $F_2$, then there exists a bivariate copula $C_{12}$
such that:

$$F_{12}(x_1, x_2) = C_{12}(F_1(x_1), F_2(x_2)) = C_{12}(u_1, u_2) \tag{1}$$

with $u_1 = F_1(x_1)$ and $u_2 = F_2(x_2)$. For more theoretical details, we refer to Sklar (1959)
and Nelsen (2006).

The use of copulas allows to decompose the construction of a joint distribution function in
two independent steps, i.e. the modelling of the dependence structure and the modelling of the
marginal distribution functions (Nelsen, 2006; Salvadori and De Michele, 2007). As such, copulas
allow the use of complex marginal distribution functions (Salvadori et al., 2007). Because of this
advantage, the application of copulas is becoming more and more popular in hydrological and
meteorological studies. However, due to the complication in the construction of the copula model
for more than two variables, most research is limited to the bivariate case (Pham et al., 2016).

A flexible construction method for high-dimensional copulas, known as the vine copula con-
struction, has been introduced in the work of Bedford and Cooke (2001, 2002), in which multivari-
ate copulas are built by decomposing the multivariate density into a product of bivariate copula
densities. Vine copulas constitute two main advantages. First, they are simple and straightfor-
ward to apply. Second, they are very flexible and have the ability to model all types of dependence
because the bivariate copulas can be selected from a wide range of copula families (Kurowicka and
Cooke, 2007; Aas et al., 2009; Czado, 2010).

There is, however, a large number of possible decompositions for the construction of vine
copulas (Aas et al., 2009); for example, there are 24 and 240 different constructions of vine copulas
for the four- and five-dimensional case, respectively (Aas et al., 2009). Examples of two regular
four-dimensional vine copulas are given in Fig. 2(a, b). One usually focuses on two special types
of regular vine copulas: Canonical vine copulas (C-vine copulas) and D-vine copulas (Kurowicka
and Cooke, 2007). If all mutual dependences involve the same variable, the construction yields a
C-vine copula (Fig. 2(c)). If all mutual dependences are considered one after the other, i.e. the
first with the second, the second with the third, the third with the fourth, etc., the construction




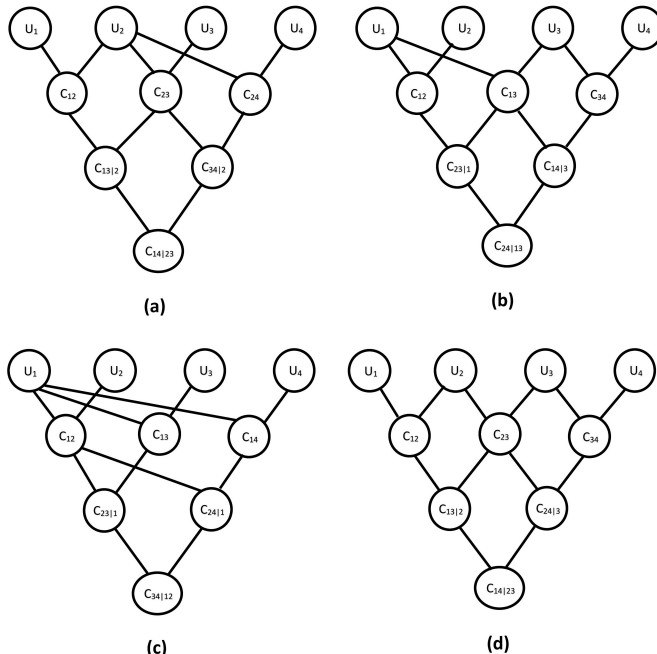

Figure 2: Examples of four-dimensional vine copulas: (a, b) regular vine copulas, (c) canonical vine or C-vine copula, (d) D-vine copula.

yields a D-vine copula (Fig. 2(d)). Sine because C-vine copulas are easier to construct than D-vine copulas, the former are selected in this study for the constructions of copula-based generators of temperature and evapotranspiration. More details of the construction and simulation from a C-vine copula are given in the work of Aas et al. (2009).

### 2.3.2 Copula-based stochastic simulation of evapotranspiration

In order to generate stochastic time series of evapotranspiration, we make use of the vine-copula-based approach proposed in the work of Pham et al. (2016) in which C-vine copulas are used to describe the dependences between evapotranspiration and other variables, such as temperature, precipitation and dry fraction within a day. The advantage of the method is that the statistical properties of the evapotranspiration time series and the dependence structures between evapotranspiration and other variables are well maintained. Furthermore, the model construction and simulation are simple to apply. After comparing the results of different vine models, Pham et al. (2016) found that the best simulations of daily evapotranspiration were provided by the four-dimensional C-vine copula $V_{TPDE}$ relating daily temperature ($T$), precipitation ($P$), dry fraction ($D$) and evapotranspiration ($E$), and the three-dimensional C-vine copula $V_{TPE}$ relating $T$, $P$ and $E$. As there is no major difference in performance between simulations using $V_{TPDE}$ and $V_{TPE}$ (Pham et al., 2016), for simplicity, we consider to use only $V_{TPE}$ in which the Frank copula family is selected for modelling the dependences between variables. A shown in (Pham et al., 2016), the White goodness-of-fit test (Schepsmeier, 2015) indicated that the Frank copula family allows for describing the dependence structure of the data included in the $V_{TPE}$. In order to avoid the seasonal effects, a different C-vine copula model is used for each month. More details on the comparison of several evapotranspiration copula-based models can be found in Pham et al. (2016).

The construction of $V_{TPE}$ is given as follows (see Fig. 3(a)). First, values $(u_{T,j}, u_{P,j}, u_{E,j})$ of





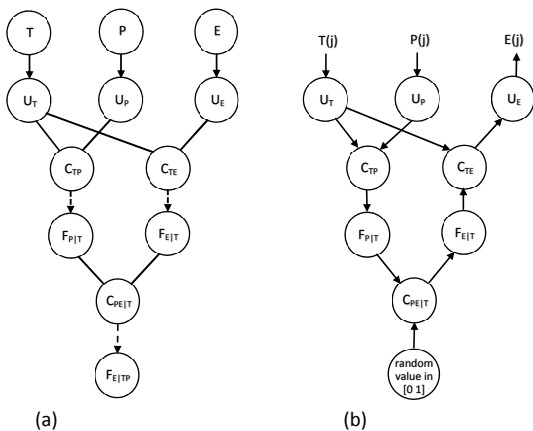

(a)  (b)

Figure 3: Construction of C-vine copula $V_{TPE}$ (a) and simulation of $E$ from $V_{TPE}$ (b)

$U_T$, $U_P$ and $U_E$ are derived from the marginal distributions of respectively $T$, $P$ and $E$ ($j = 1, ..., n$ and $n$ is the number of data points), and are used to select and fit the bivariate copulas $C_{TP}$ and $C_{TE}$, respectively. These bivariate copulas are conditioned on $U_T$ through partial differentiation, resulting in the conditional cumulative distribution functions $F_{P|T}$ and $F_{E|T}$. Using these two conditional distributions, the conditional probabilities are calculated for all data points. To these probabilities, which are also uniformly distributed on [0,1], a bivariate copula $C_{PE|T}$ is fitted, of which the partial derivative to $F_{P|T}$ can be computed to obtain $F_{E|TP}$. Once the C-vine copula model is fitted, a corresponding time series of evapotranspiration values can be generated, for a given time series of rainfall and temperature data, by sampling the copula (Fig. 3(b)). To that end, values of $U_E$ are calculated as:

$$u_E = F_{E|T}^{-1}(F_{E|TP}^{-1}(r|u_T, u_P)) \qquad (2)$$

where $r$ is a random value drawn from a uniform distribution on [0,1]. Then the corresponding evapotranspiration value $e$ can be calculated using the inverse marginal distribution function:

$$e = F_E^{-1}(u_E) \qquad (3)$$

It is clear that the values of $U_E$ are affected by the random value $r$, therefore, several simulations will show some variability. To account for these stochastic effects, the simulation was repeated 100 times. Figure 4 displays the comparisons between frequency distributions of observed and simulated evapotranspiration obtained by $V_{TPE}$ for the different months. From these plots, it can be seen that the frequency distributions of the stochastic evapotranspiration are very similar to those of the reference evapotranspiration in Uccle (red line). In order to assess whether the dependence structures between simulated evapotranspiration and other variables are maintained, for each of the 100 simulations, the mutual dependences between $E$ and the other variables, $T$ or $P$, were assessed via Kendall's tau for each month. Figure 5 shows box plots of the obtained values of Kendall's tau for $E$ vs. $T$ and $E$ vs. $P$ dependences for 100 simulations. These figures show that, in general, the observed dependences between both $E$ vs. $T$ and $E$ vs. $P$ are preserved with the stochastic simulated evapotranspiration.

### 2.3.3 Copula-based stochastic simulation of temperature

Temperature data are required for the stochastic modelling of evapotranspiration. However, in situations where no long-term time series of temperature is available, it is necessary to use a stochastically-generated temperature time series. We use a similar approach as Pham et al. (2016) to develop a stochastic temperature model based on copulas. This model makes use of





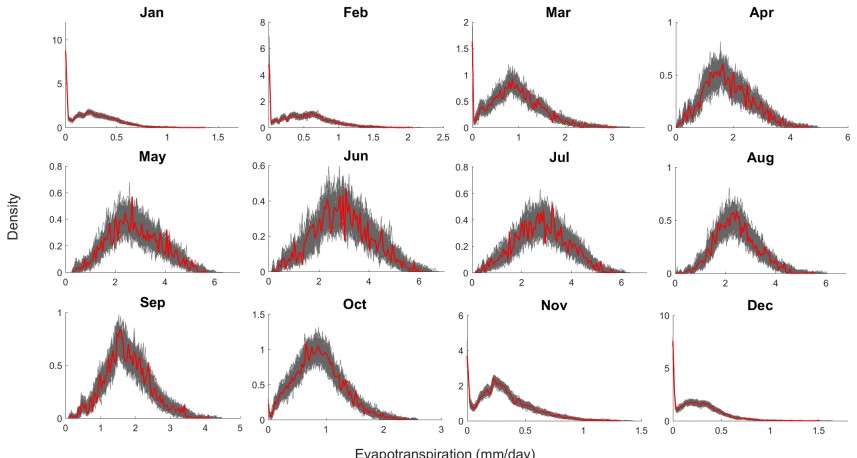

Figure 4: Comparison between the frequency distributions of evapotranspiration of observed and simulated values: Uccle (red), the ensemble of 100 time series simulated using the C-vine copula $V_{TPE}$ (grey).

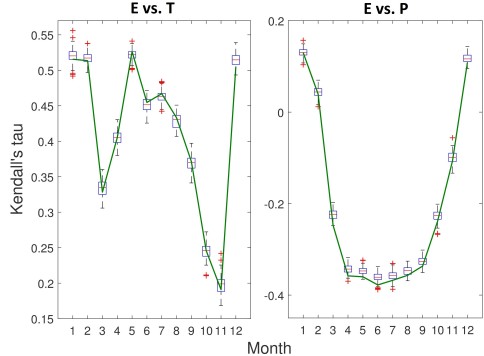

Figure 5: Comparison between Kendall's tau for the relations of $E$ vs. $T$ (left) and $E$ vs. $P$ (right) of observed and simulated values: Uccle (green line), 100 simulated time series (box plot)

the dependence between the temperature and the precipitation of the same day (i.e. at day $j$) and the temperature of the previous day (i.e. at day $j-1$). Firstly, the correlation between the temperature at day $j$ ($T_j$) and the temperature at the previous day ($T_{j-1}$) is assessed by the Pearson correlation coefficient. Given the high correlation, i.e. 0.94, we thus can conclude that there is a strong dependence between $T_j$ and $T_{j-1}$. Similarly as for the stochastic evapotranspiration model, a C-vine copula is employed in which $T_{j-1}$ is chosen as the core variable. The model is referred to as $V_{T_p PT}$, where $T_p$ refers to the temperature of the previous day.

The construction procedure of $V_{T_p PT}$ is similar to the one of $V_{TPE}$ (see Section 2.3.2). The simulation process of the temperature model is different from that of the evapotranspiration model, in the sense that it requires a modelled input from the previous time step (i.e. $T_p$) in order to generate a new value for $T$. The simulation algorithm of $T$ can be performed as follows:




Table 1: Bivariate copulas selected by AIC for $V_{T_pPT}$, where F stands for Frank, Ga for Gaussian, G for Gumbel, C for Clayton and J for Joe

| Month | $V_{T_pPT}$ | | |
| --- | --- | --- | --- |
| | $c_{T_pP}$ | $c_{T_pT}$ | $c_{PT|T_p}$ |
| Jan | F | Ga | F |
| Feb | F | Ga | Ga |
| Mar | F | Ga | F |
| Apr | F | Ga | F |
| May | F | Ga | F |
| Jun | F | Ga | F |
| Jul | F | Ga | F |
| Aug | F | Ga | F |
| Sep | F | Ga | Ga |
| Oct | C | Ga | Ga |
| Nov | C | Ga | F |
| Dec | F | Ga | F |

$$u_T = F_{T|T_p}^{-1}(F_{T|T_pP}^{-1}(r|u_{T_p}, u_P)) \tag{4}$$

$$t = F_T^{-1}(u_T) \tag{5}$$

In order to maintain the dependence structures between variables, but still keep the model simple and easy to construct, the best bivariate copulas for the C-vine copula are chosen using the Akaike's information criterion (AIC) (Akaike, 1973) from five one-parameter copula families, i.e. the Gaussian, the Clayton, the Gumbel, the Frank and the Joe family. Table 1 illustrates which copulas were selected. This table shows that the Frank copula family is often selected for $C_{T_pP}$ and $C_{PT|T_p}$, while the Gaussian copula is often chosen for $C_{T_pT}$. To keep the copula-based simulation procedure simple, we restrict the model to use only a combination of Frank-Gaussian-Frank for the C-vine copula $V_{T_pPT}$. Further, the White goodness-of-fit test (Schepsmeier, 2015) is applied to check whether the dependence present in the data is captured by the Frank-Gaussian-Frank C-vine copulas. With $p$-values larger than 0.05 for all months, we find that the dependence structure of the data can be described by the selected copulas. These copulas are then used for generating temperature given the time series of precipitation.

To assess the performance of the model, the statistics of 100 stochastic time series of temperature using the observed daily precipitation from 1931 to 2002 are compared to those of the observations. The empirical cumulative distribution functions (ECDF) of the monthly mean temperature for each of the simulated 72-year time series are shown in Fig. 6. The statistics of the simulations seem to be relatively similar to the observations. Figure 7 shows the monthly maximum temperature of the ensemble and of the observed temperature series corresponding to empirical return periods. This figure shows that the extremes are well modelled for all months.





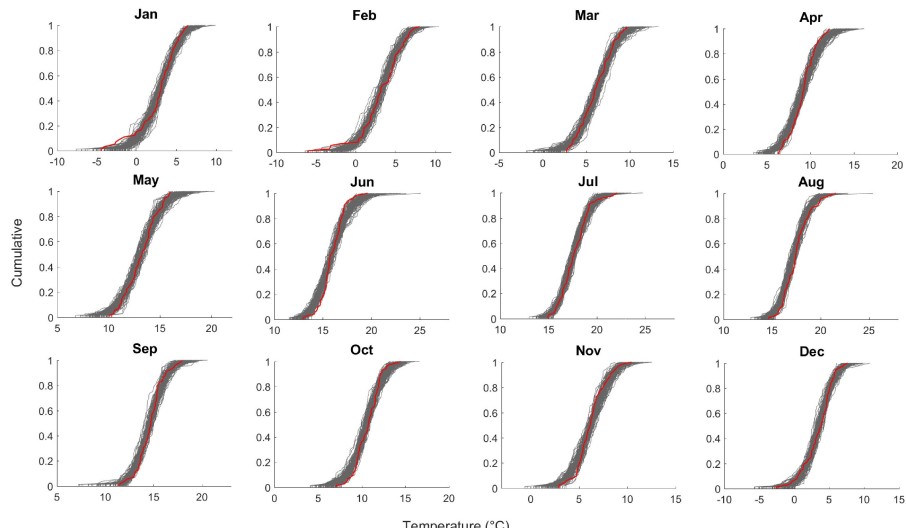

Figure 6: Comparison between the empirical cumulative distribution function (ECDF) of the monthly mean $T$ of the observed and simulated values: Uccle (red), the ensemble of 100 time series simulated using the C-vine copula $V_{TpPT}$ (grey).

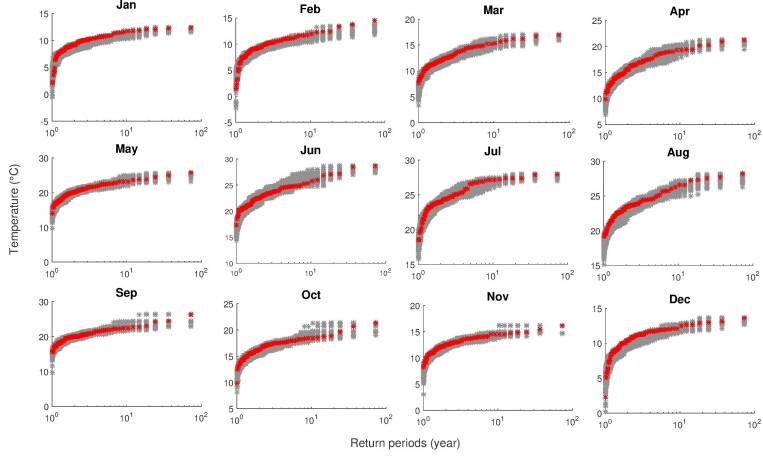

Figure 7: Comparison between the return periods of monthly extremes of the observed and simulated temperature values: Uccle (red), the ensemble of 100 time series simulated using the C-vine copula $V_{TpPT}$ (grey).





Table 2: Optimal parameter set for the (monthly) MBL model.

| Parameter | $\lambda$ | $\kappa$ | $\phi$ | $\mu_x$ | $\alpha$ | $\nu$ |
|---|---|---|---|---|---|---|
| January | 0.021 | 0.009 | 0.002 | 11.037 | 12.042 | 0.833 |
| February | 0.014 | 0.008 | 0.001 | 15.000 | 4.041 | 0.143 |
| March | 0.018 | 0.009 | 0.001 | 15.000 | 5.393 | 0.219 |
| April | 0.017 | 0.151 | 0.032 | 0.823 | 20.000 | 19.029 |
| May | 0.023 | 1.130 | 1.000 | 0.371 | 4.000 | 14.420 |
| June | 0.016 | 0.089 | 0.059 | 1.190 | 10.064 | 20.000 |
| July | 0.012 | 0.012 | 0.004 | 7.676 | 20.000 | 5.715 |
| August | 0.010 | 0.003 | 0.001 | 15.000 | 19.963 | 2.729 |
| September | 0.014 | 0.199 | 0.100 | 0.417 | 4.000 | 14.039 |
| October | 0.013 | 8.949 | 0.096 | 0.095 | 4.000 | 2.488 |
| November | 0.023 | 0.121 | 0.026 | 1.061 | 4.000 | 2.486 |
| December | 0.014 | 0.005 | 0.001 | 14.998 | 20.000 | 1.792 |

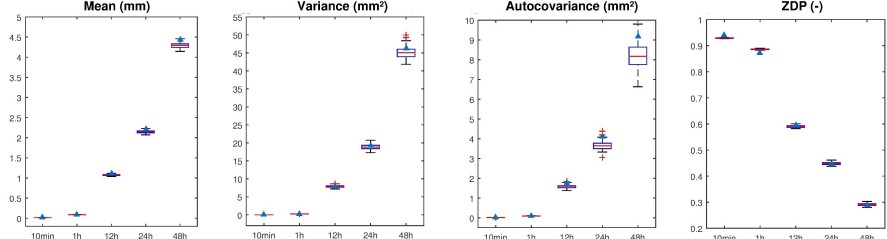

Figure 8: Comparison between observed and simulated precipitation data for the mean, variance, auto-covariance and zero-depth probability (ZDP): Uccle (blue triangle), the ensemble of 100 simulated time series by the MBL model (box plot).

## 2.4 Simulated precipitation by the MBL model

In situations where no long time series of precipitation is available, one can use a stochastic rainfall model. In this study, the modified Bartlett–Lewis (MBL) model (Rodriguez-Iturbe et al., 1988) is selected to generate the precipitation time series based on the results from Pham et al. (2013) in which the MBL model is considered to be the best version of the different BL models tested on the Uccle data set. The MBL model is calibrated based on the mean, variance, lag-1 autocovariance and zero-depth probability (ZDP) at the aggregation levels of 24 h, 48 h and 72 h instead of 10 min, 1 h and 24 h that were used in Pham et al. (2013). The reason for only selecting aggregation levels of at least one day is to consider situations where only daily precipitation data would be available. The values of the calibrated parameters are given in Table 2. Details of the MBL model and the model calibration are provided by Pham et al. (2013). The stochastic rainfall time series is simulated at the same 10-minute time resolution as the observations. In order to assess the performance of the model, the abilities of the model to reproduce some general historical statistics, such as mean, variance, the lag-1 autocovariance and ZDP, at aggregation levels of 10 min, 1 h, 12 h, 24 h and 48 h are investigated based on an ensemble of 100 time series.

In Fig. 8, some general statistics at different aggregation levels are compared for 100 time series obtained by the MBL model and the observed time series in Uccle. In order to further unveil the behaviour of the model, the general statistics are calculated at different aggregation levels for each year and presented in the form of an ECDF (Fig. 9). From both figures, it can be seen that the mean is generally reproduced well by the model at all levels of aggregation. At the sub-hourly level, the variance and autocovariance are slightly overestimated. For higher aggregation levels, an increasing variation is found for both statistical properties. At higher levels of aggregation, the ZDP is similar to that found for the observed time series, whereas for hourly and sub-hourly



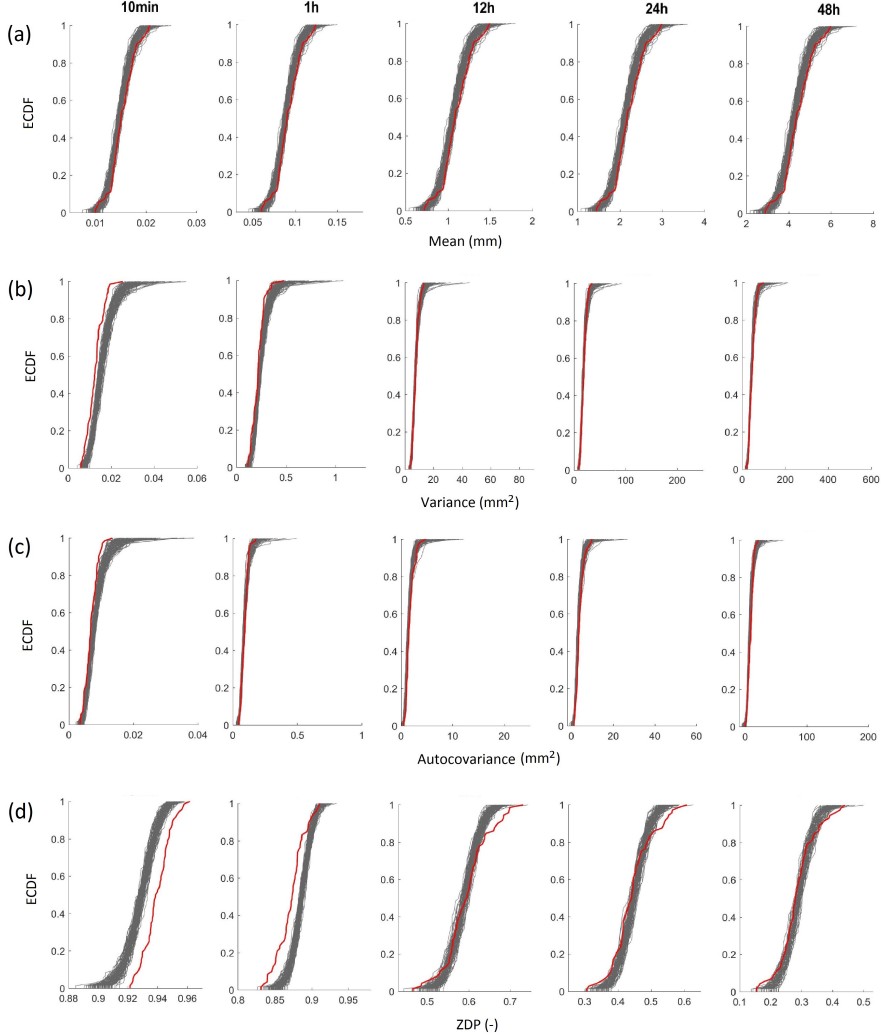

Figure 9: Comparisons between the empirical cumulative distribution function of mean, variance, autocovariance and ZDP calculated for the observed and simulated precipitation data for different aggregation levels for each year: Uccle (red), 100 simulated time series by the MBL model (grey). ECDFs are shown for the (a) mean, (b) variance, (c) lag-1 autocovariance and (d) the zero-depth probability (ZDP).





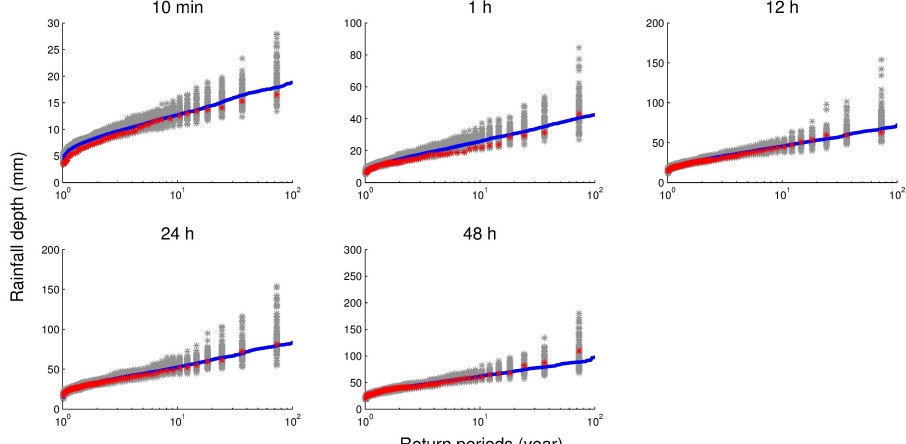

Figure 10: Comparisons between the return periods of extremes of the observed and simulated precipitation data at different aggregation levels: Uccle (red), the ensemble of 100 simulated time series by the MBL model (grey). Calculation of the extremes for a given return period on a time series that is based on concatenating the 100 simulated time series, results in the blue line

levels, a slight deviation in ZDP-values are found with respect to the observations.

Figure 10 shows the empirical univariate return periods of the annual maximum rainfall depths of the observed and simulated series, considering five different aggregation levels. Compared to the observations, it seems that the MBL model is able to preserve the maxima at all aggregation levels. It can be seen in this study that the MBL model does not suffer from the problem of underestimation of extreme values at sub-hourly aggregation levels that were reported in the work of Verhoest et al. (1997) and Cameron et al. (2000). From the analysis, it seems that the MBL model is capable of preserving the sub-daily statistics even though the calibration procedure only included daily and multi-day statistics. Yet, further research is needed for exploring this improved behavior.

Figure 10 also shows that a large variation in extreme values is found for larger return periods. The MBL model allows for generating rainfall time series mimicking the statistics of the observed series. Due to its structure, the modeled precipitation values are not restricted to the range of rainfall values in the observations, making this model able to generate rainfall events having a return period larger than the observed time series. Yet, it can thus be expected that within the modeled time series of 72 years, events may occur having a true return period that is larger than the length of the modeled time series. If longer time series would be simulated, a better estimation of the rainfall corresponding to return periods that are smaller than the observed time series should be obtained. To demonstrate this, all 50 series generated are concatenated, resulting in one time series of $50 \times 72 = 3600$ years, for which the return periods are calculated empirically and plotted (only for return periods less then 100 years) as a blue line in Figure 10. As can be seen for return periods smaller than 100 years, a good fit with the observations are obtained, showing that MBL is capable of reproducing extremes. Yet, the user should use much longer time series than the maximum return period aimed for.

# 3    Discharge simulation scenarios

The catchment discharge is calculated by the PDM that uses precipitation and evapotranspiration data as inputs. In order to assess the impact of each stochastic variable on the modelling of discharge, three cases have been developed that can be compared to a reference situation (cfr. Fig. 11).





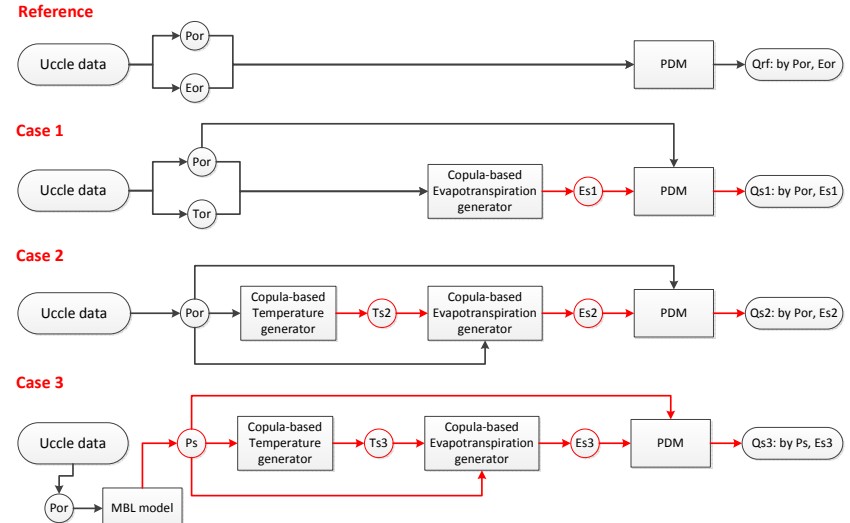

Figure 11: Different cases for discharge simulation. $P_{or}$, $E_{or}$ and $T_{or}$ refer to the observed time series. $P_s$, $E_{s1}$, $E_{s2}$, $E_{s3}$, $T_{s2}$ and $T_{s3}$ refer to the simulated time series (red block). Red arrows indicate the simulation processes related to stochastically-generated time series.

The reference situation is obtained by running the PDM with the observed time series of precipitation and evapotranspiration. In case 1, it is supposed that insufficient evapotranspiration data would be available (e.g. a shorter time series than the observed precipitation), the stochastic evapotranspiration can then be generated using the three-dimensional C-vine copula, i.e. $V_{TPE}$, given observed rainfall and temperature. The simulation is repeated 50 times in order to account for stochastic effects. In case 2, where only a sufficient long time series of precipitation is available, the process starts with temperature simulations, then evapotranspiration can be modelled using the observed precipitation and stochastically-generated temperature using the $V_{TPE}$ copula. As presented before, temperature values will be generated by the three-dimensional C-vine copula $V_{T_pPT}$ that relates temperature $T$ to daily precipitation $P$ and the daily temperature of the previous day $T_p$. To account for stochastic effect, 50 time series of temperature are generated. Next, each of 50 time series of temperature, together with the observed precipitation data, are used to simulate 50 corresponding time series of evapotranspiration. Therefore, in total 2500 time series of evapotranspiration are generated. Case 3 accounts for a situation in which data would insufficiently be available for all input variables. In this case, an ensemble of 50 time series of precipitation could be generated using the MBL model. For each of these time series, 50 time series of temperature and 2500 time series of evapotranspiration can be obtained using the same approach in case 2. In total, 125000 time series of evapotranspiration are generated in case 3. In order to construct copula models and evaluate discharge simulations in all cases, this study uses the same time series of precipitation, evapotranspiration and temperature at Uccle. In all cases, discharge is simulated using the PDM that was calibrated for the Grote Nete catchment in Belgium (see Section 2.2). By this approach, the uncertainty due to the PDM can be partly excluded from the study, i.e. we study the change in performance with respect to the reference situation. It makes sense because the three cases use exactly the same PDM, a similar uncertainty due to the model is assumed for all cases as for the reference situation. Therefore, the change in performance for all cases with respect to the reference situation can be attributed to the differences in inputs to the model. The discharge simulations in the three cases are denoted as $Q_{s1}$, $Q_{s2}$ and $Q_{s3}$, respectively, while the reference discharge is denoted by $Q_{rf}$.





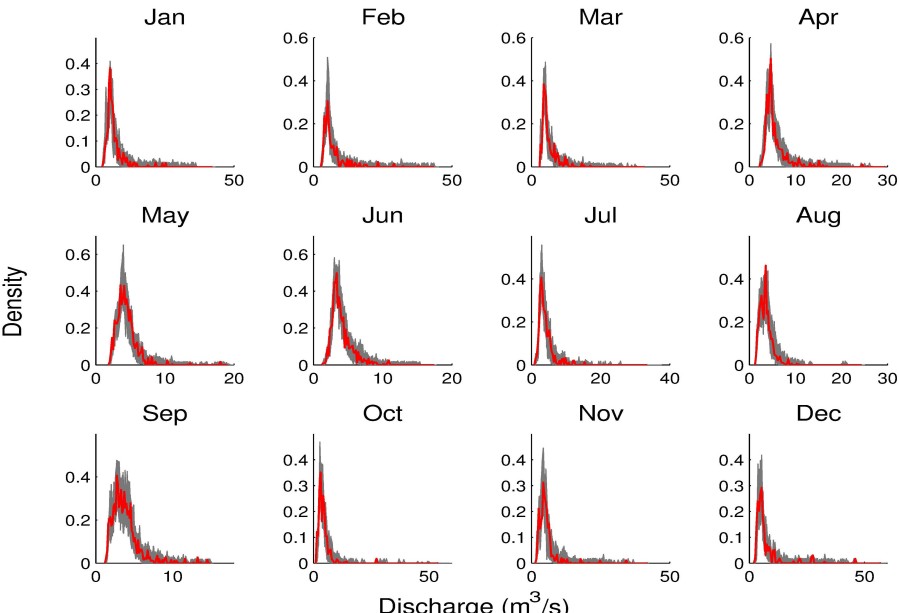

Figure 12: Comparison between the frequency distributions of the reference discharge $Q_{rf}$ (red) and the ensemble of 50 time series of discharge values simulated using observed precipitation and simulated evapotranspiration values in case 1 (grey).

## 4 Results and discussions

### 4.1 Case 1

The catchment discharge can be simulated by means of the PDM that uses precipitation and evapotranspiration data. In case 1 (cfr. Fig. 11), where only daily observed precipitation and temperature data are available, 50 stochastically-generated evapotranspiration time series are generated using the three-dimensional C-vine copula $V_{TPE}$. The results shown in Section 2.3.2 and the work of Pham et al. (2016) reflect that the C-vine copula $V_{TPE}$ performs well and its simulations lie very close to the values of the observed evapotranspiration. Figure 12 displays the comparison between the frequency distributions of $Q_{rf}$ and $Q_{s1}$ for the different months. It can be seen that the distributions of $Q_{s1}$ are quite similar to those of the reference discharge for all months. For a further analysis of mean discharges and annual extremes of $Q_{s1}$, we refer to Section 4.3.

### 4.2 Case 2

In case 2 (cfr. Fig. 11), only a time series of precipitation of sufficient length is available and the temperature values are simulated using the C-vine copula $V_{T_pPT}$. The observed precipitation and stochastically-generated temperature values are then used for reproducing the evapotranspiration by means of the C-vine copula $V_{TPE}$. Through comparing the results of this case with that of case 1, we can assess the impact of introducing a stochastic temperature model on the modelled evapotranspiration time series and the modelled discharge.

As shown in Section 2.3.3 and Fig. 13, the stochastically-generated temperature data generated by the C-vine copula $V_{TpPT}$ model are reliable and can be used together with the recorded precipitation to simulate 2500 time series of evapotranspiration in the next step (i.e. for each tem-





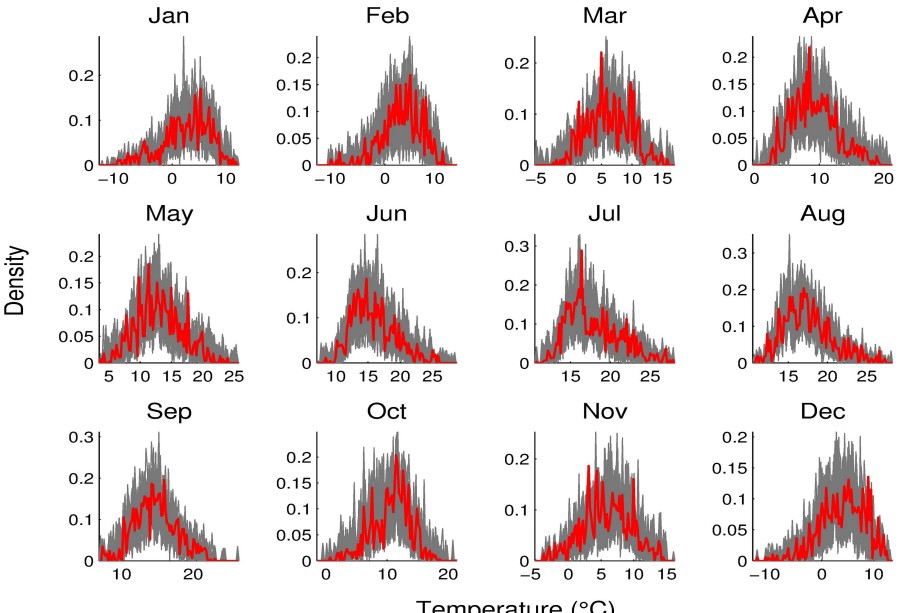

Figure 13: Comparison between the frequency distributions of temperature of the observed and simulated values in case 2: Uccle (red), the ensemble of 50 time series simulated using the C-vine copula $V_{T_pPT}$ (grey).

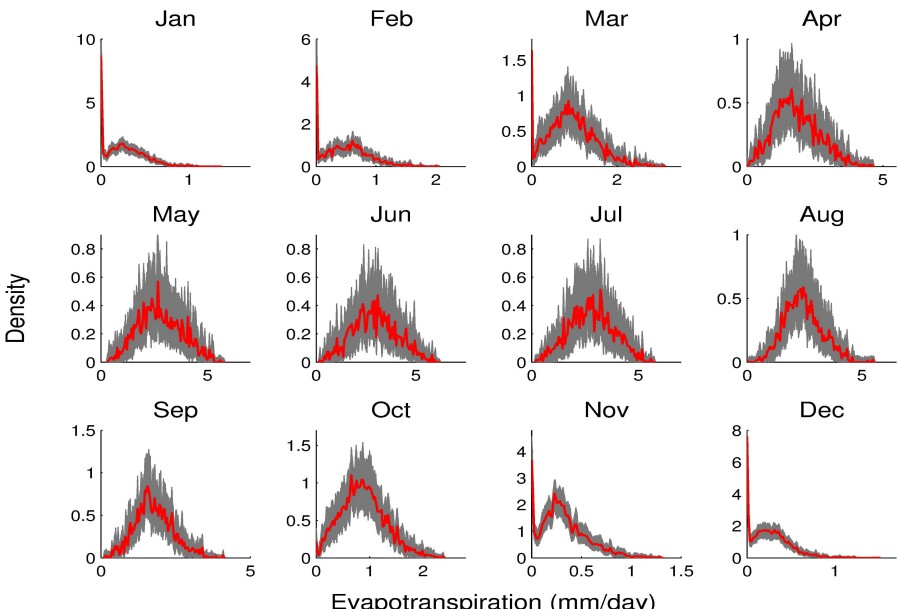

Figure 14: Comparison between the frequency distributions of evapotranspiration of the observed and simulated values in case 2: Uccle (red), the ensemble of 2500 time series simulated using the C-vine copula $V_{TPE}$ (grey).





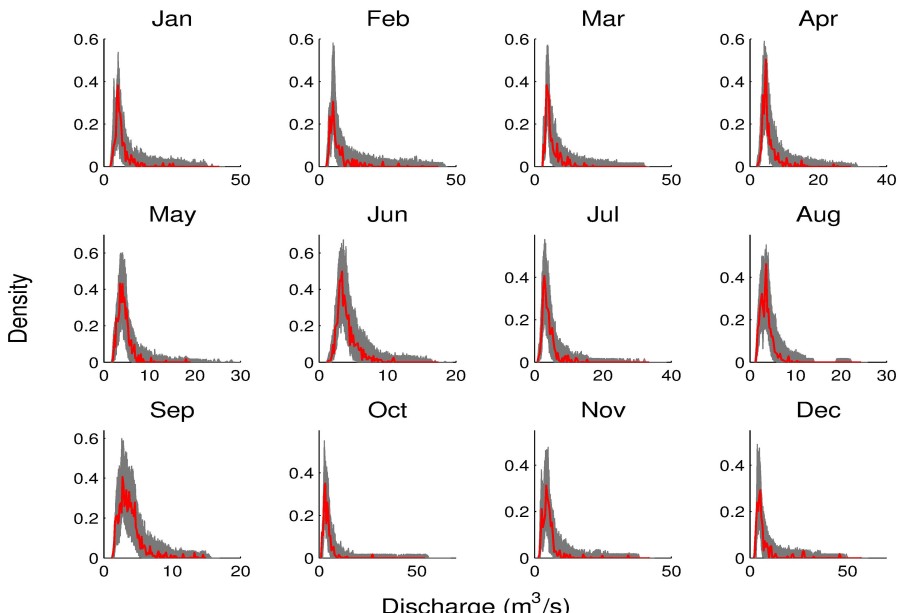

Figure 15: Comparison between the frequency distributions of reference discharge (red) and the ensemble of 2500 time series of discharge values simulated using the observed precipitation and simulated evapotranspiration in case 2 (grey).

perature series, 50 evapotranspiration series are generated). The frequency distributions of the
2500 time series of the simulated evapotranspiration are shown in Fig. 14. It can be seen from the
figures that these distributions are similar to those of the observations in Uccle and those of the
modelled evapotranspiration in case 1 for all months. Figure 15 displays a comparison between
the frequency distributions of the simulated discharge ($Q_{s2}$) and the reference discharge ($Q_{rf}$).
In general, the grey areas representing 2500 simulated time series are slightly wider than those in
case 1. We conclude that the introduction of stochastically-generated temperature does not cause
considerable deviations in the simulation of evapotranspiration and discharge.

## 4.3  Case 3

This case accounts for a situation in which no time series (of sufficient length) are available as
shown in Fig. 11. The first step consists of generating 50 time series of precipitation by means of
the MBL model (see Section 2.4) and aggregating these to the daily level. Then, each of those time
series is used for modelling 50 time series of temperature, each used for generating 50 evapotranspi-
ration series. Therefore, in total 125000 time series of evapotranspiration are generated. Finally,
125000 time series of the catchment discharge are simulated using the stochastically-generated
time series of precipitation and corresponding evapotranspiration values. This case will allow for
assessing the uncertainty introduced by using the MBL model for generating precipitation values
as input to a rainfall-runoff model.

First, the simulated time series of precipitation are used as inputs to the C-vine copula $V_{T_pPT}$
to generate time series of temperature. The modelled copula-based temperature values are com-
pared with the observed temperature in Uccle in terms of the frequency distributions in Fig. 16.
From these figures, it can again be seen that the distributions of the simulations follow those of
the observations. With respect to the frequency distributions, the simulated evapotranspiration
(Fig. 17) in this case is similar to the observed evapotranspiration, but more deviations can be





observed in this case than in the previous cases. The modelled time series of precipitation and evapotranspiration are then used for modelling the discharge. The frequency distributions of the simulated discharge values for the different months are displayed in Fig. 18. From the different plots, it can be concluded that the simulations still follow the distribution of the reference discharge (red line).

Compared to the simulated discharge of cases 1 and 2, more higher extreme values are generated and the grey areas representing the ensemble of 125000 time series are generally wider, indicating that mainly the stochastic generation of precipitation has introduced considerable variations into the discharge simulations. This increase in uncertainty should however be treated with care. As stated before, the generated rainfall series may include extremes that are larger than the ones in the observed time series. Such large precipitation values will inevitably result in a large surface runoff production causing extreme discharges. The large variability in extreme rainfall as observed in Figure 10 will consequently lead to large variabilities in modeled extreme discharges (cfr. Figure 19). If, however, the discharge extremes from a longer time series are studied, the variation in extremes is strongly reduced. To demonstrate this, 50 rainfall time series of 3600 year and corresponding evapotranspiration time series (remark that only one series is generated per rainfall time series) are used as input to the rainfall-runoff model, and the extremes, having return periods smaller than 500 years, are plotted for each of these 50 time series (Figure 20). As can be seen, the large uncertainties in extremes, encountered when using 72 year time series as input, are highly reduced, showing a slight overestimation for larger return periods, if compared to those modeled using the observed time series of rainfall and evapotranspiration. Yet, it is impossible to state whether true overestimations are obtained, or that, due to the stochastic nature or rainfall (and evapotranspiration), the observations used never resulted in extreme discharge events that actually exceed a (true) 40-year return period event (i.e., the maximum discharge based on the observed time series of precipitation and evapotranspiration corresponds to a return period of about 40 years based on the simulations using the modelled very long time series of precipiation and evaporation). Similarly as discussed for Figure 10, this result makes a plea for using modeled discharge time series of a length that is a multiple of the maximum return period of discharge aimed at, where longer time series reduce the variation in discharge values at high return periods at the expense of run-time. Further research will be needed to seek for the trade-off between length of the time series and the remaining uncertainty.





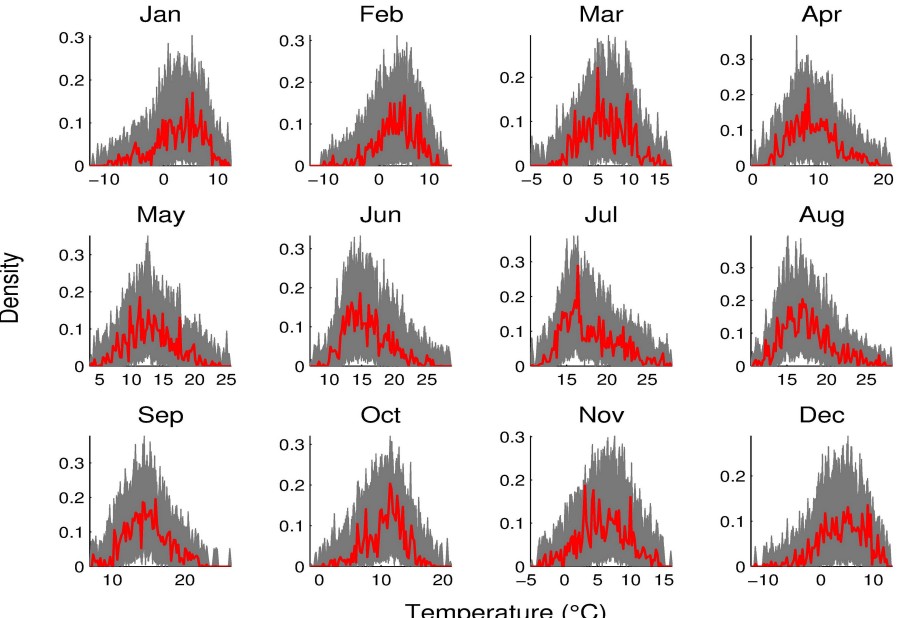

Figure 16: Comparison between the frequency distributions of temperature of the observed and simulated values in case 3: Uccle (red), the ensemble of 2500 time series simulated using the C-vine copula $V_{TpPT}$ (grey).

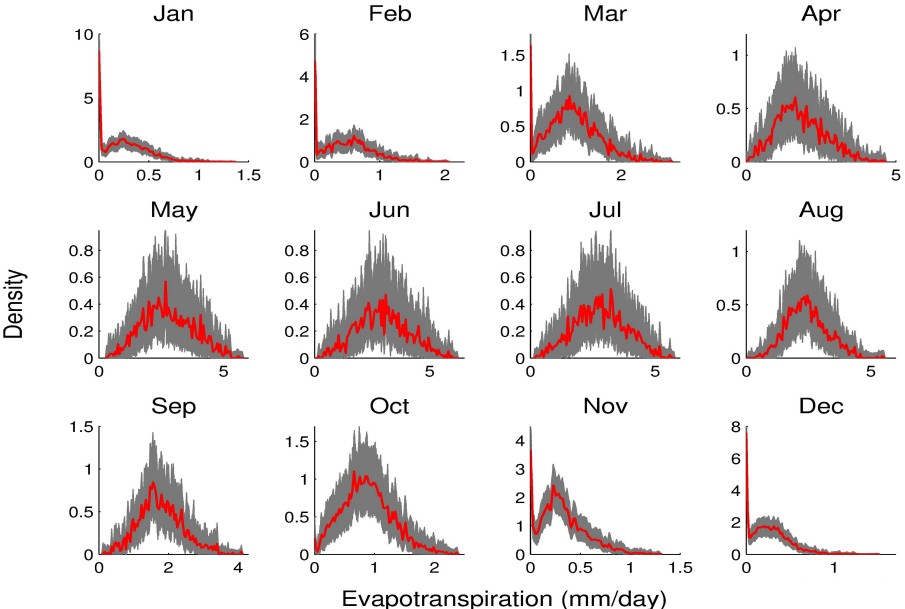

Figure 17: Comparison between the frequency distributions of evapotranspiration of the observed and simulated values in case 3: Uccle (red), the ensemble of 125000 time series simulated using the C-vine copula $V_{TPE}$ (grey).





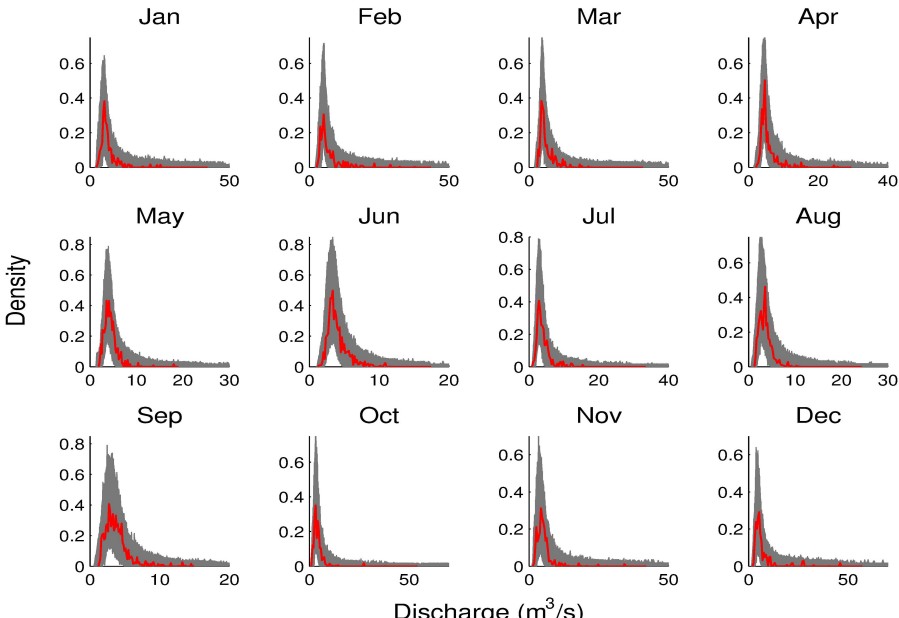

Figure 18: Comparison between the frequency distributions of reference discharge $Q_{rf}$ (red) and the ensemble of 125000 time series of discharge values simulated using the simulated precipitation and evapotranspiration in case 3 (grey).

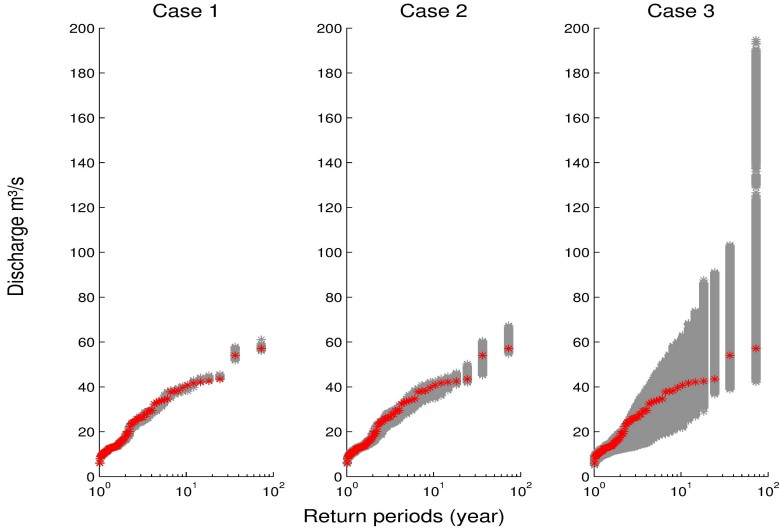

Figure 19: Comparison between the empirical return periods of annual extremes of the observed and simulated discharge for all cases: reference discharge $Q_{rf}$ (red), the ensemble of time series of simulated discharge (grey).





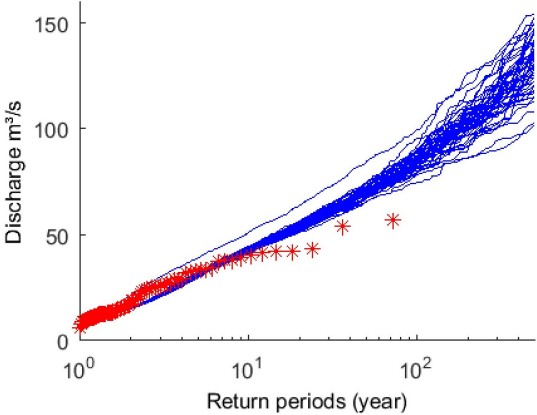

Figure 20: Comparison between the empirical return periods of annual extremes of the observed and simulated discharge for case 3 based on 50 time series of 3600 years of rainfall and corresponding evapotranspiration.

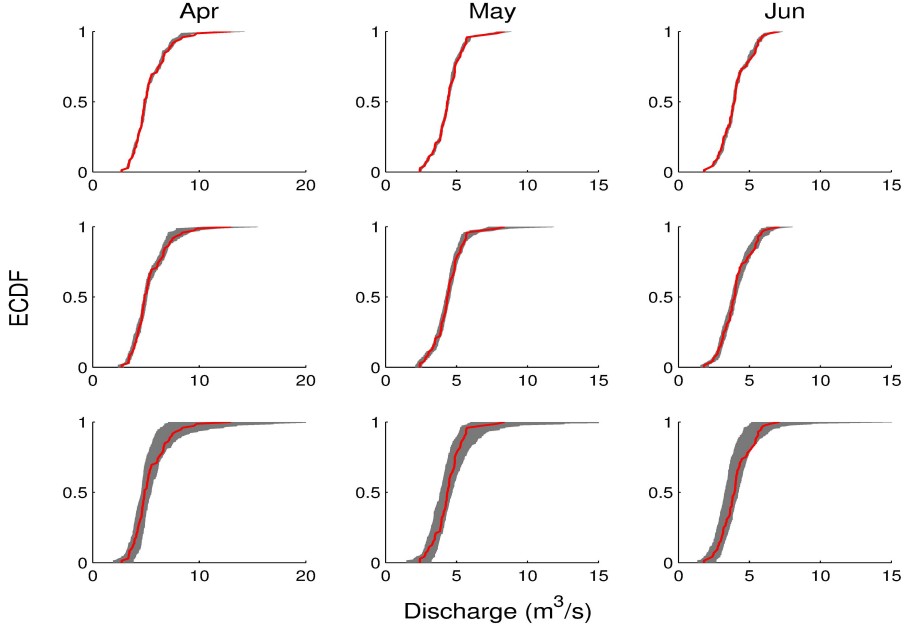

Figure 21: Comparison between the ECDF of the mean of discharge for Apr - Jun of the observed and simulated values in three cases: reference discharge $Q_{rf}$ (red), the time series of simulated discharge (grey).





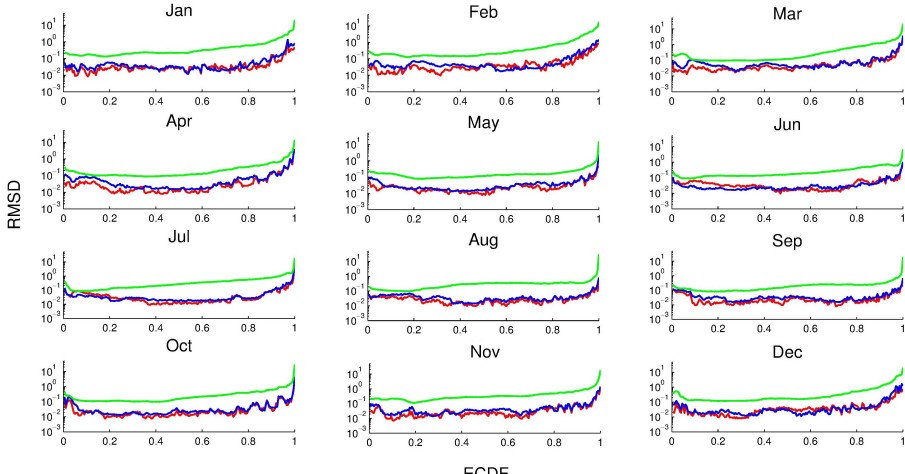

Figure 22: Root mean square difference (RMSD) for simulated discharge in different cases: case 1 (red), case 2 (blue) and case 3 (green).

In order to further investigate the quality of the simulated discharge for all cases, Fig. 21 presents the comparison between the ECDF of the daily averages of the modelled and reference discharge for April, May and June. For all cases, the daily mean seems to be preserved by the modelled discharge. However through investigating the width of the grey areas of the simulated time series for each case, as expected, we can conclude that the most certain results are observed in case 1, followed by case 2 and case 3. This also holds for the other months. Similar situations are witnessed for the univariate return period of annual extreme discharge (Fig. 19) in which the least and largest variations between the reference and simulated discharge are noticed for $Q_{s1}$ and $Q_{s3}$, respectively. Especially, a remarkable expansion of grey areas is witnessed in case 3. It is clear that each stochastic component, i.e. modelled precipitation, temperature or evapotranspiration, has contributed an additional amount of variation to the modelled discharge. The differences between the simulated discharge from different cases are less evident in terms of frequency distributions but more pronounced for the mean and extreme discharge.

To account for the variations between the modelled and reference discharge, the simulated discharge values are further evaluated using the root mean square deviation (RMSD):

$$\text{RMSD}(i) = \sqrt{\frac{1}{n}\sum_{s=1}^{n}\left(Q_{m,s}(i) - Q_o(i)\right)^2} \qquad (6)$$

where $Q_m(i)$ and $Q_o(i)$ are respectively the modelled and reference discharge value that have the same value of cumulative frequency $i \in [0,1]$, $i = 0.005, ..., 1$ with a step of 0.005; and $n$ is the number of the members in the ensemble considered.

Figure 22 displays the RMSD calculated for simulated discharge in different cases. It can be seen from the figure that for all cases, larger RMSD values are found for the higher values of discharge. In other words, simulations of the higher values of discharge are generally less accurate. There are insignificant differences between the RMSD for case 1 and 2 for all months. The use of stochastically-generated temperature time series seemed to contribute minor uncertainty to the discharge simulations in this study. The largest errors often are obtained in case 3 where the discharge is simulated from stochastically-generated precipitation and evapotranspiration values.





## 5    Conclusions

In water management, discharge is a very important variable which can be simulated via a rainfall-runoff model using recorded precipitation and evapotranspiration data. However, in situations that suffer from data deficiency, one may consider using stochastically-generated time series. In this study, the impact of using the stochastically-generated precipitation and evapotranspiration on the simulation of the catchment discharge is investigated. In order to assess the influence of each stochastic variable on the discharge simulations, three different cases have been considered. In the first case, it is assumed that insufficient evapotranspiration data would be available, requiring stochastically-generated evapotranspiration based on observed precipitation and temperature data by means of a copula. In the second case, where only precipitation data would be sufficiently available, the temperature and evapotranspiration are each reproduced by vine copulas. The third case addresses the situation where too short time series of observations are available. In this case, the precipitation time series could be generated using a Modified Bartlett-Lewis (MBL) model calibrated to the limited precipitation data available and then the time series of temperature and evapotranspiration could be obtained using the copula-based models. In all cases, the C-vine copulas $V_{TPE}$ and $V_{T_pPT}$ are used for the simulations of evapotranspiration and temperature, respectively. From the comparison between the simulations with the observations, the C-vine copulas seem to reproduce the time series of evapotranspiration and temperature well. It is clear that each stochastic component has a certain impact on the discharge simulations, and each additional stochastic variable will contribute an additional variation, and thus uncertainty. As expected, the simulations of the discharge obtained for case 1 show the smallest variability, while those in case 3 results in the largest variability. In general, no major differences are observed between the simulations and observations in cases 1 and 2, the characteristics of the discharge series seem to be preserved through the process for these cases. Noticeable variations are witnessed in case 3, where the discharge is simulated using modeled time series of precipitation and evapotranspiration.

With respect to extreme discharge, it was shown that the uncertainties encountered in case 3 are highly reduced when using much longer time series as input than the maximum return period aimed at. However, given that all forcing data are generated, the modeller is not restricted to the length of an observed time series, but can generate time series of whatever length as input to the hydrological model, taking into account that the longer the time series used, the more the uncertainty reduces at the expense of increasing run-time.

From this study, we may thus conclude that in situations that suffer from a lack of observations, one can rely on the stochastically-generated series of precipitation, temperature and evapotranspiration to reproduce time series of discharge for water resources management. However, care should be taken as the modelled extreme discharges may experience the largest errors.

## Acknowledgements

The authors gratefully acknowledge the Vietnamese Government Scholarship (VGS), the King Baudouin Foundation (KBF) and the project G.0013.11N of the Research Foundation Flanders (FWO) for their partial financial support for this work. The historical Uccle series were provided by the Royal Meteorological Institute of Belgium.

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
