# Peer review of "A coupled stochastic rainfall-evapotranspiration model for hydrological impact analysis"

_Hydrology and Earth System Sciences, 2017_

## Referee Comment (RC1) · T. Nagler (Referee) · 16 Jun 2017

**Comment on "A coupled stochastic rainfall-evapotranspiration model for hydrological impact analysis" by Minh Tu Pham et al.**

Thomas Nagler

June 16, 2017

**1 General comments**

The manuscript is very well written and gives sufficient context to understand the relevant developments and issues in hydrological impact analysis. The authors clearly motivate why a stochastic rainfall-evapotranspiration model is useful in this context. Their proposal is based on vine copulas, a modern statistical tool for modeling stochastic dependence between multiple variables. This is a laudable effort, but the way this methodology is applied and its performance is evaluated is problematic in several ways. I fully acknowledge that HESS is not a statistics journal and statistical subtleties may not matter in specific applications. But the extent to which they do in this particular context are unclear and needs to be addressed.

Below I identify three major issues and explain why they are problematic from a statistical perspective. I urge the authors to thoroughly evaluate the implications for their hydrological model. Where possible, I try to make suggestions for alternative methods that may improve their model and its assessment. Since the first two issues may be equally relevant for other readers, my comments will be more elaborate than what is common in a closed review.

**2 Specific comments**

**2.1 Major issues**

**2.1.1 Seasonal effects**

A copula models the dependence between two random variables $X_1, X_2$ with marginal distributions $F_1$ and $F_2$. Its parameters can be estimated from observations of these variables, $\boldsymbol{X}_t = (X_{1,t}, X_{1,t})$, $t = 1, \ldots, T$. The usual assumption for the validity of the estimate is that the data $\boldsymbol{X}_t$ are independent and identically distributed (*iid*). In particular, the distribution of $\boldsymbol{X}_t$ should not change with $t$, which is usually violated by climatic variables. The authors acknowledge that by fitting multiple models, one for each month.

I am afraid this may not be enough, because climatic trends also exist within months. For example: in central Europe, the end of April is — on average — much warmer than the beginning of April. Suppose that additionally, the average

precipitation is decreasing during April. Then high temperatures will likely coincide with low levels of precipitation and vice versa. A copula fitted to this data will show negative dependence, which merely reflects the two deterministic within-month trends working in opposite directions, but not the stochastic dependence between the time series.

Whether or not within-month trends exist can be easily checked visually or by formal statistical tests (e.g., Harris and Sollis, 2003, Chapter 3). If they do exist, they should be accounted for on a finer time scale. Splitting the data into weeks or even days could be a solution, but significantly decreases the number of observations available for fitting the copula model. A good alternative is to center and scale the time series by their seasonal mean and standard deviation. More specifically, if $X_{j,d,y}$ denotes variable $j$ observed at day $d$ of year $y$, set

$$\tilde{X}_{j,d,y} = (X_{j,d,y} - \mu_{j,d})/\sigma_{j,d}, \tag{1}$$

where $\mu_{j,d}$ and $\sigma_{j,d}$ are the mean and standard deviation of $X_{j,d,y}$, $y = 1, \ldots, 72$. If necessary, trends in the skewness of $\tilde{X}_{j,d,y}$ can be removed similarly using a Box-Cox transformation. This transformation is usually sufficient to account for deterministic seasonal effects. We can now build a copula for the stochastic dependence in $\tilde{\boldsymbol{X}}_t = \tilde{\boldsymbol{X}}_{d+365(y-1)}$. Simulated data from this model can be transformed to the original scale by inverting (1).

**2.1.2 Inter-serial dependence**

Even when the distribution of $\tilde{X}_t$ is the same for each $t$, subsequent observations of the time series may not be independent. Such data is called *stationary* which is less restrictive than *iid*. Typically, stationarity is sufficient to allow for valid estimation of the marginal distributions and copula of $\tilde{X}_t$. But inference tools (like confidence intervals and goodness-of-fit tests) derived under the *iid* assumption are no longer valid.

Another potential issue is that inter-serial dynamics can play an important role in applications. If so, these dynamics should be modeled explicitly explicitly. In the context of hydrological discharges, this is likely the case. Large discharges often occur when extreme weather conditions have been persistent for several days, and persistence is a sign of inter-serial dependence. A simple way to check whether such dependence is present is to look at the autocorrelation of the time-series, i.e., the correlation between $\tilde{X}_t$ and $\tilde{X}_{t-1}$ (and their squares). If the correlation is small, one can test statistically whether it is zero.

If there is dependence, there are two popular ways to capture it:

1. **Copula models**: This route is taken by the authors in 2.3.3, but only for the temperature variable. Similar models for the inter-serial dependence in evapotranspiration and precipitation should be employed in addition. If $F_{j,t,t-1}$ is the joint distribution of $\tilde{X}_{j,t}$ and $\tilde{X}_{j,t-1}$, the between-variables dependence can be modeled by a copula for the variables

$$U_{1,t} = F_{1,t|t-1}(\tilde{X}_{1,t} \mid \tilde{X}_{1,t-1}), \quad U_{2,t} = F_{2,t|t-1}(\tilde{X}_{2,t} \mid \tilde{X}_{2,t-1}),$$

where $F_{j,t|t-1}$ is the conditional distribution of $\tilde{X}_{j,t}$ given $\tilde{X}_{j,t-1}$.

2. **Classical time series models**: Classical time series models (see, e.g., Shumway et al., 2000, Chapter 3) assume that the variable $\tilde{X}_{j,t}$ is a linear combination of the preceding values ($t' < t$) and *iid* noise. For example, the autoregressive model of order $p$ is

$$\tilde{X}_{j,t} = \sum_{k=1}^{p} \phi_{j,k} \tilde{X}_{j,t-k} + \epsilon_{j,t},$$

where $\phi_{j,k}$ are model parameters and $\epsilon_{j,t}$ is *iid* noise with mean zero. The sequence $\epsilon_{j,t}$ is commonly called *innovation* or *residual* series. The stochastic between-variable dependence can then be captured by a copula model for $(\epsilon_{1,t}, \epsilon_{2,t})$. More complex models are required when $\tilde{X}_{j,t}^2$ is autocorrelated (Harris and Sollis, 2003, Chatper 8).

**2.1.3   Assessing the quality of the vine copula model**

There are multiple issues with how the quality of the model is evaluated:

1. To check the model's validity, the authors merely look at the density/cdf of the observed and simulated values of a single time series. This is only weakly related to the vine copula model and not a good indicator for its fit. Under this measure, just simulating from the distribution $F_E$ (thereby assuming that $E$ is independent of $T$ and $P$) would lead to results that are at least as good as the ones from the vine copula.

   To adequately assess the quality of the dependence model, pair-wise comparisons should be made. For example, one can look at the scatter plots of observed and simulated pairs $(X_{1,t}, X_{2,t})$. Another alternative are contour plots of the estimated joint density of observed vs. simulated pairs. Such comparisons should be made for all variable combinations. Similarly, multivariate return periods should be considered instead of single-variable return periods (see, Salvadori et al., 2011).

2. Figures 6 and 9 use empirical cumulative distribution functions (ECDF) instead of densities for no obvious reason. I advise against using ECDF's because they suggest a misleading sense of closeness between distributions. Since ECDFs are necessarily monotone functions with boundary values 0 and 1, their shape is quite restricted. For example, the left panels of Figure 9(d) show that the distributions are different, but the ECDFS still look somewhat similar. But the corresponding densities would show almost no overlap and more clearly communicate the dissimilarity.

3. In Section 4, the uncertainty in the simulation model is assessed for various degrees of data availability. From the spread of estimated densities in Figures 11-15, the authors conclude that uncertainty increases when a variable is not observed and needs to be simulated. This is likely true, but can not be inferred from these figures. The density plots for cases 1-3 are based on a different number of observations. The spread seen on 125000 simulations will naturally be larger than the the spread on 50 observations — even when the actual distribution is the same. Hence, the spreads should only be compared when they are based on the same number of simulations.

**2.2   Minor issues**

1. Since vine copulas are the essential ingredient in your model, I suggest to indicate this in the abstract.

2. p. 4, p. 165: If unconditional bivariate copulas are used (as is common), a vine copula is not a decomposition, but a construction. A decomposition is called *non-simplified vine copula* and involves conditional bivariate copulas (see, e.g., Stöber et al., 2013). I suggest to rephrase this sentence.

3. p. 4, l. 167: I suggest to change "all types of dependence" to "a wide range of dependence structures". "All types" can only be modeled by a non-simplified vine copula.

4. p. 5, l. 179: What do you mean by "C-vine copulas are easier to construct than D-vine copulas"? In fact, any three-dimensional vine is both a C- and D-vine, which can be easily verified by re-arranging the vertices of the vine graph.

5. p. 6, l. 203 ff.: I am afraid a reader without prior knowledge of vine copulas will not understand your paragraph on how the model is estimated. Instead of your explanation, it should suffice to refer the reader to Aas et al. (2009).

6. How are marginal distributions modeled/estimated?

7. Figures 4, 12–17 should use a larger smoothing parameter to decrease variability of the density estimates. A large proportion of the observed variability is due to the density estimation technique. This is not the kind of variability you want to assess.

8. Figure 22: What is $i$?

**3   Technical corrections**

1. p. 5, l. 179: "Sine because C-vine ..." should be "Because C-vine ...".

**References**

Aas, K., Czado, C., Frigessi, A., and Bakken, H. (2009). Pair-copula constructions of multiple dependence. *Insurance: Mathematics and economics*, 44(2):182–198.

Harris, R. and Sollis, R. (2003). *Applied time series modelling and forecasting*. Wiley.

Salvadori, G., De Michele, C., and Durante, F. (2011). On the return period and design in a multivariate framework. *Hydrology and Earth System Sciences*, 15(11):3293–3305.

Shumway, R. H., Stoffer, D. S., and Stoffer, D. S. (2000). *Time series analysis and its applications*, volume 3. Springer.

Stöber, J., Joe, H., and Czado, C. (2013). Simplified pair copula constructions—limitations and extensions. *Journal of Multivariate Analysis*, 119:101–118.

---

## Referee Comment (RC2) · M. Sadegh (Referee) · 13 Jul 2017

**Review comments on "A coupled stochastic rainfall-evapotranspiration model for hydrological impact analysis" by Pham et al. 2017 submitted to HESSD**

**Mojtaba Sadegh**

This is a nice study that uses copula-based approaches to stochastically generate mutually dependent rainfall and evapotranspiration forcing for rainfall runoff models. This could be used for design purposes where observation is sparse or missing. However, this approach does not seem to be working properly for the extreme events (which are needed for design purposes). Acknowledging this fact, the study is valuable for the areas with no observation.

Overall, paper is well written and well structured. I have some comments (most of them major) that could potentially improve the quality of this paper.

1. Line 46: Authors use stochastic process models to generate precipitation series. My question is:
   How do stochastic process models handle changing characteristics of precipitation? Several studies have shown, for parts of the world, that rainfall events are shrinking in time and expanding in amplitude. Also there is a temporal shift in rainfall events in some parts of the world, let alone the changes in the distribution of rainfall/snow. Addressing these issues could be helpful.

2. Lines 91-94: I don't understand how the number of stochastically generated forcing data could influence the uncertainty of the rainfall-runoff model's response. Uncertainty is a characteristic of the forcing data (let's neglect the modeling uncertainties for now), not the number of generated time series. So if you find a time series that fit your runoff extremes well, this is just a random phenomenon. This cannot be the basis for prediction, as we can't determine the best forcing for future, and need to rely on the ensemble of forcing data.

3. Lines 95-96: Section 2 should precede section 3!

4. I am confused about how sections 2.1 and 2.2 are connected. Historical record of climate forcing are obtained for Brussels, and RR model is calibrated for the Grote Nete catchment. How do you use a model calibrated against one watershed, to predict runoff at another watershed?
   Moreover, 1 year of data for evaluation is not enough. You will need a couple of years to ensure calibrated model can capture different aspects of a catchment.

5. Section 2.3: Copulas characterize dependence structure of different variables. This means there should be a dependence structure. Did you quantify the correlation between evaporation, temperature, and precip? If so, is it significant? At what temporal scale?

My understanding is that you perform your analysis at daily scale, and I fear the correlation might not be significant at the daily scale.

6. Line 150: bivariate -- > It could be multivariate

7. Line 152: I would reference to Joe 1997 too. Joe and Nelsen both played an important role in introducing copula to the scientific community.

8. Lines 171-173: I agree that vine copulas are very flexible, but it comes at a price! A model with 4 degrees of freedom is more flexible than a competitor with 2! However, usually there is not enough information to constrain all parameters. The copula literature usually does not address the parameter uncertainties, and so they neglect the identifiability of parameters. I would address this predicament here. For more info, refer to figure 6 of:
Sadegh, M., E. Ragno, and A. AghaKouchak (2017), Multivariate Copula Analysis Toolbox (MvCAT): Describing dependence and underlying uncertainty using a Bayesian framework, Water Resour. Res., 53, doi:10.1002/2016WR020242.
Link: http://onlinelibrary.wiley.com/doi/10.1002/2016WR020242/full

9. Line 191: As a minor issue, when someone talks about a 3-dimensional model, I expect the model to have three parameters. When someone talk about trivariate model, I expect a multivariate model that associates three variables.

10.      Line 203: How did you construct the marginal distribution? Empirical? Fitted distribution?

11.      Eq. 3: how did you calculate inverse of the vine copula? Analytical or numerical?

12.      Line 253: pvalue larger than 0.05 or smaller?!

13.      Section 2.4: How did you calibrate the modified Bartlett–Lewis (MBL) model, given the stochastic nature of precipitation prediction models? With stochastic models, usually summary statistics of data and simulation are compared, rather than original time series. For this purpose, approximate Bayesian computation is a great framework.

14.      Lines 338-344: I cannot disagree more! Forcing and model uncertainties are intertwined, and interact in a nonlinear manner. It is not as simple as you explained. You cannot simply use a RR model calibrated for one watershed to simulate runoff at another watershed! Tens (Hundreds) of papers are available on the regionalization topic, not many of them really provided a sound ground for transferring model parameters from one watershed to another! Worse is that authors assume this modeling uncertainty does not interact with the forcing uncertainty.

15.      Figure 22: 22 figures? Is that many figures really necessary when most of them don't provide any new info?

16.      Lines 476-478: I have a hard time accepting this claim. If you generate a much longer synthetic (stochastic) forcing, then let's say predictions at a 100 years return period level improves. I accept this. But I cannot accept the general comment that longer fording data reduces overall uncertainties. What if I had to estimate a 500 years return period flow?

---

## Author Comment (AC1) · 11 Sep 2017

**1 General comments of the first reviewer**

The manuscript is very well written and gives sufficient context to understand the relevant developments and issues in hydrological impact analysis. The authors clearly motivate why a stochastic rainfall-evapotranspiration model is useful in this context. Their proposal is based on vine copulas, a modern statistical tool for modeling stochastic dependence between multiple variables. This is a laudable effort, but the way this methodology is applied and its performance is evaluated is problematic in several ways. I fully acknowledge that HESS is not a statistics journal and statistical subtleties may not matter in specific applications. But the extent to which they do in this particular context are unclear and needs to be addressed. Below I identify three major issues and explain why they are problematic from a statistical perspective. I urge the authors to thoroughly evaluate the implications for their hydrological model. Where possible, I try to make suggestions for alternative methods that may improve their model and its assessment. Since the first two issues may be equally relevant for other readers, my comments will be more elaborate than what is common in a closed review.

*We would like to thank the referee for his valuable comments and extensive explanation of the statistical background. We hereby list the comments of the referee and formulate our answers in italics.*

**2 Specific comments of the first reviewer**

**2.1 Major issues**

**2.1.1 Seasonal effects**

A copula models the dependence between two random variables $X_1$, $X_2$ with marginal distributions $F_1$ and $F_2$. Its parameters can be estimated from observations of these variables, $\boldsymbol{X}_t = (X_{1,t}, X_{2,t})$, $t = 1, \ldots, T$. The usual assumption for the validity of the estimate is that the data $\boldsymbol{X}_t$ are independent and identically distributed (iid). In particular, the distribution of $\boldsymbol{X}_t$ should not change with $t$, which is usually violated by climatic variables. The authors acknowledge that by fitting multiple models, one for each month. I am afraid this may not be enough, because climatic trends also exist within months. For example: in central Europe, the end of April is on average much warmer than the beginning of April. Suppose that additionally, the average precipitation is decreasing during April. Then high

temperatures will likely coincide with low levels of precipitation and vice versa. A copula fitted to this data will show negative dependence, which merely reflects the two deterministic within-month trends working in opposite directions, but not the stochastic dependence between the time series. Whether or not within-month trends exist can be easily checked visually or by formal statistical tests (e.g., Harris and Sollis, 2003, Chapter 3). If they do exist, they should be accounted for on a finer time scale. Splitting the data into weeks or even days could be a solution, but significantly decreases the number of observations available for fitting the copula model. A good alternative is to center and scale the time series by their seasonal mean and standard deviation. More specifically, if $X_{j,d,y}$ denotes variable $j$ observed at day $d$ of year $y$, set

$$\tilde{X}_{j,d,y} = (X_{j,d,y} - \mu_{j,d})/\sigma_{j,d}\,, \tag{1}$$

where $\mu_{j,d}$ and $\sigma_{j,d}$ are the mean and standard deviation of $X_{j,d,y}$ , $y = 1,\dots,72$. If necessary, trends in the skewness of $\tilde{X}_{j,d,y}$ can be removed similarly using a Box-Cox transformation. This transformation is usually sufficient to account for deterministic seasonal effects. We can now build a copula for the stochastic dependence in $\tilde{\boldsymbol{X}}_t = \tilde{\boldsymbol{X}}_{d+365(y1)}$ . Simulated data from this model can be transformed to the original scale by inverting (1).

*We thank the referee for this elaborative explanation. On the basis of his comments, we checked for each variable and each month for the existence of within-month trends. For each month and each variable, we therefore first visualised the data using box plots, each containing the 72 observations for each day of the month, cf. we have a 72-yearly time series. Figure 2.1.1 illustrates these box plots for temperature, precipitaion and evapotranspiration for the month March.*

*This figure clearly demonstrates the existence of a within-month trend for temperature and evapotranspiration. This was also confirmed by an ANOVA test, employed when distributions were homoscedastic, a Welch ANOVA test, employed when distributions were heteroscedastic or a Kruskal Wallis test, employed when distributions were not-normal and heteroscedastic. For these tests the significance level $\alpha$ was set to 0.001. For precipitation no trend was observed, which was also confirmed by the results of the statistical tests. Therefore, we decided to standardize the temperature and evapotranspiration data as suggested by the referee. For each month, vine copula models will then be built on the basis of standardized temperature and evapotranspiration data and non-standardized precipitation data.*

[Figure]

Figure 1: Boxplots per day of the month for temperature (top left), precipitation (top right) and evapotranspiration (bottom) in March

**2.1.2 Inter-serial dependence**

Even when the distribution of $\tilde{X}_t$ is the same for each $t$, subsequent observations of the time series may not be independent. Such data is called stationary which is less restrictive than *iid*. Typically, stationarity is sufficient to allow for valid estimation of the marginal distributions and copula of $\tilde{X}_t$. But inference tools (like confidence intervals and goodness-of-fit tests) derived under the *iid* assumption are no longer valid. Another potential issue is that inter-serial dynamics can play an important role in applications. If so, these dynamics should be modeled explicitly. In the context of hydrological discharges, this is likely the case. Large discharges often occur when extreme weather conditions have been persistent for several days, and persistence is a sign of inter-serial dependence. A simple way to check whether such dependence is present is to look at the autocorrelation of the time-series, i.e., the correlation between $\tilde{X}_t$ and $\tilde{X}_{t1}$ (and their squares). If the correlation is small, one can test statistically whether it is zero. If there is dependence, there are two popular ways to capture it:

1. **Copula models**: This route is taken by the authors in 2.3.3, but only for the temperature variable. Similar models for the inter-serial dependence in evapotranspiration and precipitation should be employed in addition. If $F_{j,t,t1}$ is the joint distribution of $\tilde{X}_{j,t}$ and $\tilde{X}_{j,t1}$, the between-variables dependence can be modeled by a copula for the variables

$$U_{1,t} = F_{1,t|t1}(\tilde{X}_{1,t}|\tilde{X}_{1,t1}), \quad U_{2,t} = F_{2,t|t1}(\tilde{X}_{2,t}|\tilde{X}_{2,t1}), \qquad (2)$$

where $F_{j,t|t1}$ is the conditional distribution of $\tilde{X}_{j,t}$ given $\tilde{X}_{j,t1}$.

2. **Classical time series models**: Classical time series models (see, e.g., Shumway et al., 2000, Chapter 3) assume that the variable $\tilde{X}_{j,t}$ is a linear combination of the preceding values ($t' < t$) and *iid* noise. For example, the autoregressive model of order $p$ is

$$\tilde{X}_{j,t} = \sum_{k=1}^{p} \phi_{j,k}\tilde{X}_{j,tk} + \epsilon_{j,t}, \qquad (3)$$

where $\phi_{j,k}$ are model parameters and $\epsilon_{j,t}$ is *iid* noise with mean zero. The sequence $\epsilon_{j,t}$ is commonly called innovation or residual series. The stochastic between-variable dependence can then be captured by a copula model for $(\epsilon_{1,t}, \epsilon_{2,t})$ More complex models are required when $\tilde{X}_{j,t}$ is autocorrelated (Harris and Sollis, 2003, Chapter 8).

*Again, we thank the referee for this elaborative explanation. We statistically checked for the existence of autocorrelation on the monthly time series by employing a Ljung-Box Q-test with a significance level $\alpha$ of 0.001. As the total time series covers 72 years of data, this test was performed for each month and repeated 72 times. On this basis we found that autocorrelation exists for temperature and evapotranspiration. We will first take this autocorrelation into account by extending the vine $V_{TPE}$ with $E_{t-1}$. In a further stage, we will couple the vines $V_{T_pPT}$ and $V_{TPE_pE}$, where $T_p$ and $E_p$ denote temperature and evapotranspiration of the previous day, by fitting a copula between the conditioned values $F_{T|T_pP}$ and $F_{E|TPE_p}$. This coupling will then be taken into account in the sampling procedure as to generate coupled time series of temperature and evapotranspiration.*

**2.1.3 Assessing the quality of the vine copula model**

There are multiple issues with how the quality of the model is evaluated:

1. To check the models validity, the authors merely look at the density/cdf of the observed and simulated values of a single time series. This is only weakly related to the vine copula model and not a good indicator for its fit. Under this measure, just simulating from the distribution $F_E$ (thereby assuming that $E$ is independent of $T$ and $P$ ) would lead to results that are at least as good as the ones from the vine copula. To adequately assess the quality of the dependence model, pair-wise comparisons should be made. For example, one can look at the scatter plots of observed and simulated pairs $(X_{1,t}, X_{2,t})$. Another alternative are contour plots of the estimated joint density of observed vs. simulated pairs. Such comparisons should be made for all variable combinations. Similarly, multivariate return periods should be considered instead of single-variable return periods (see, Salvadori et al., 2011).

   *The vine copula models are employed to generate time series of temperature or evapotranspiration that reflect the properties (statistics) of the original observed time series. To this end, different time series were generated by the vine copula models and their distributions were compared to the distribution of the observed time series. Furthermore, in order to check whether or not the dependence between the generated variables is maintained, the values of Kendall's tau were examined (see Figure 5 of the original manuscript). By examining scatter plots be-*

*tween observed and simulated pairs, as the referee suggests, one would rather investigate whether the simulated values approximate the observed values, as in a prediction model. In this study, however, it is not the aim to match the observed values for each time step t as closely as possible, but rather mimic the behaviour of the observed time series w.r.t. statistics and extremes.*

2. Figures 6 and 9 use empirical cumulative distribution functions (ECDF) instead of densities for no obvious reason. I advise against using ECDFs because they suggest a misleading sense of closeness between distributions. Since ECDFs are necessarily monotone functions with boundary values 0 and 1, their shape is quite restricted. For example, the left panels of Figure 9(d) show that the distributions are different, but the ECDFS still look somewhat similar. But the corresponding densities would show almost no overlap and more clearly communicate the dissimilarity.

   *It is not clear to us what the referee means with his reference to the left panels of Figure 9(d). These panels clearly show that the distributions (the ECDFs) of the observation and the simulation differ. The left panel shows that the ZDP values of the simulations underestimate the ZDP values of the original time series, the second panel from the left shows the opposite. By showing the probability density distributions instead of the cumulative distributions, one would also see this under- or overestimation. (See Figure 2). Yet, we will change the figures and use densities instead of ECDFs.*

3. In Section 4, the uncertainty in the simulation model is assessed for various degrees of data availability. From the spread of estimated densities in Figures 11-15, the authors conclude that uncertainty increases when a variable is not observed and needs to be simulated. This is likely true, but can not be inferred from these figures. The density plots for cases 1-3 are based on a different number of observations. The spread seen on 125000 simulations will naturally be larger than the the spread on 50 observations even when the actual distribution is the same. Hence, the spreads should only be compared when they are based on the same number of simulations.

[Figure]

[Figure]

We are aware of the fact that the density plots for cases 1-3 are based on a different number of observations. This is due to the fact that in case 1 one stochastic model is employed, and hence "only" 50 time series have to be generated, whereas two or three stochastic models are employed in respectively case 2 and 3. In the build-up of this manuscript, at first we "only" generated 50 time series of evapotranspiration in all cases. However, this means that in cases 2 and 3, only one time series of temperature respectively temperature and precipitation was generated. In such a setting, the stochasticity of $V_{TpPT}$ and the $MBL$ model would be ignored. For this reason, we decided to also generate 50 time series of temperature in case 2 and for each of these time series 50 time series of evapotranspiration. In case 3, we hence generated 50 time series of precipitation and for each of these time series, we generated 50 time series of temperature. For each of the generated temperature time series, we also generated 50 time series of evapotranspiration conform case 1. We are aware that this increases the spread, which is also indicated in the paper, but we prefer not to ignore the stochasticity of the other employed models in cases 2 and 3.

In order to compare the spreads on the basis of the same number of simulations, we will carry out the following excercise. We will also generate for one time series of P and T, 50 time series of E and compare these separately w.r.t. the spread. By repeating this several times, we can hence compare the spread on the basis of the same number of observations and still take into account the stochasticity.

**2.2 Minor issues**

1. Since vine copulas are the essential ingredient in your model, I suggest to indicate this in the abstract.

   *We will change the abstract to better stress this.*

2. p. 4, p. 165: If unconditional bivariate copulas are used (as is common), a vine copula is not a decomposition, but a construction. A decomposition is called non-simplified vine copula and involves conditional bivariate copulas (see, e.g., Stöber et al., 2013). I suggest to rephrase this sentence.

   *We will rephrase this sentence.*

3. p. 4, l. 167: I suggest to change all types of dependence to a wide range of dependence structures. All types can only be modeled by a non-simplified vine copula.

   *This will be changed in the manuscript.*

4. p. 5, l. 179: What do you mean by C-vine copulas are easier to construct than D-vine copulas? In fact, any three-dimensional vine is both a C- and D-vine, which can be easily verified by re-arranging the vertices of the vine graph

   *This sentence will be removed from the manuscript.*

5. p. 6, l. 203 ff.: I am afraid a reader without prior knowledge of vine copulas will not understand your paragraph on how the model is estimated. Instead of your explanation, it should suffice to refer the reader to Aas et al. (2009).

   *As we think it's important to explain to the reader how we handled the estimation of the vine, we will revise this paragraph to make it more clear to the reader and add a reference to Aas et al. (2009).*

6. How are marginal distributions modeled/estimated?

   *Empirical distributions are employed as marginal distributions. This will be added in the manuscript.*

7. Figures 4, 1217 should use a larger smoothing parameter to decrease variability of the density estimates. A large proportion of the observed variability is due to the density estimation technique. This is not the kind of variability you want to assess.

*We will investigate the impact of the smoothing parameter to decrease the variability.*

8. Figure 22: What is i?

   *The values of the ECDF, indicated in the abscissa, reflect the values of i.*

**3 Comments of the second reviewer**

This is a nice study that uses copula-based approaches to stochastically generate mutually dependent rainfall and evapotranspiration forcing for rainfall runoff models. This could be used for design purposes where observation is sparse or missing. However, this approach does not seem to be working properly for the extreme events (which are needed for design purposes). Acknowledging this fact, the study is valuable for the areas with no observation. Overall, paper is well written and well structured. I have some comments (most of them major) that could potentially improve the quality of this paper.

*We thank the reviewer for his appreciation for our work and the valuable comments made. Yet, the suggestion made by the reviewer that this study would be applicable for areas without observations is not correct, as the model is fully constructed on simultaneously observed time series of precipitation, temperature and evapotranspiration.*

1. Line 46: Authors use stochastic process models to generate precipitation series. My question is:
   How do stochastic process models handle changing characteristics of precipitation? Several studies have shown, for parts of the world, that rainfall events are shrinking in time and expanding in amplitude. Also there is a temporal shift in rainfall events in some parts of the world, let alone the changes in the distribution of rainfall/snow. Addressing these issues could be helpful.

   *Two possible (and jointly occurring) phenomena may cause temporal changes in precipitation characteristics (but also of temperature and evapotranspiration). The first is the annual cycle, the second a long-term trend due to e.g. climate change. In our paper, we only account for the annual cycle by building models at a monthly basis. The second process (i.e. long-term changes) is not accounted for and is not the*

*aim of this study. We do not aim at constructing very long time series that reflect both phenomena. We mainly aim at building a model that allows for constructing alternative time series having the same characteristics as those of the current observations (for which it was shown that no long-term temporal trend is present). This model allows for constructing much longer time series than the observations, allowing, to a certain extent, to assess extreme events with a return period larger than the length of the original time series. This model also allows, as is shown in the paper, to generate alternative time series of the same length of the observed time series (i.e. the model predictions and the observed time series are likely outcomes of the meteorological process the observations are just one realisation, the model mimicks alternative realisations through a fully data-driven/statistical framework). When these time series are used as input to a hydrologic model, their outputs will provide alternative realisations resulting from the same meteorological process.*

*Given the fact that the precipitation-evaporation (P-ET) model that is developed in this paper is fully based on observed time series, we believe that the issue of temporal shifts in rainfall events in some parts of the world is not relevant. Also the changes in distributions of rainfall/snow is not accounted for. The model is only considering rainfall, as in the observed time series in Uccle, the fraction of snow events is very small. If discerning between snow and rainfall would be needed, the model should be extended.*

*We will improve the text to make sure that the reader is aware that no long-term temporal changes are considered, nor that the model can be applied in areas without observations.*

2. Lines 91-94: I don't understand how the number of stochastically generated forcing data could influence the uncertainty of the rainfall-runoff model's response. Uncertainty is a characteristic of the forcing data (let's neglect the modeling uncertainties for now), not the number of generated time series. So if you find a time series that fit your runoff extremes well, this is just a random phenomenon. This cannot be the basis for prediction, as we can't determine the best forcing for future, and need to rely on the ensemble of forcing data.

*The number of stochastically generated forcing data does not influence the uncertainty of the rainfall-runoff (RR) models response. Yet, the idea is that the actual observation time series is only one realisation*

*of the meteorological process, and therefore, the output, i.e. discharge, is also but one realisation. Yet, suppose, due to chaos apparent in the climatological system, an alternative observation time series would have occurred, then an alternative discharge record would have been obtained, different from the actual observed one. The latter will provide other design values (i.e. discharge values corresponding to a given return period) than the actual observed one. Yet, both are realisations of the same process, causing that the discharge values for a given return period actually follow a certain distribution, and thus results in uncertainty on e.g. extreme values, average values. This uncertainty cannot be assessed without evaluating an ensemble of possible outcomes of the meteorological process.*

*We will add some text to the manuscript to better describe the source of the uncertainty.*

3. Lines 95-96: Section 2 should precede section 3!

   *This will be corrected.*

4. I am confused about how sections 2.1 and 2.2 are connected. Historical record of climate forcing are obtained for Brussels, and RR model is calibrated for the Grote Nete catchment. How do you use a model calibrated against one watershed, to predict runoff at another watershed?

   *The climate forcing data obtained at Uccle are very representative for the climate observed in the Grote Nete catchment (the distance between Uccle (which is a town near Brussels, not a catchment) and the Grote Nete catchment is between 50 and 100 km). The RR model of the Grote Nete catchment was actually calibrated using data (however much shorter in length) observed near the catchment. Yet, given that the meteorological conditions are nearly the same, one can assume that the statistics of the modelled discharge obtained with the forcing data observed near the catchment and those observed at Uccle are negligible (actually, this was shown in studies performed for the Flemish Environmental Agency). Still, in this study, we compare the modelled discharge, obtained from the observations in Uccle, with forcings resulting from the P-ET model. Given the fact that the model is parameterised using the data of Uccle, the results using observed and modelled forcings can be compared. The idea of the paper is to show the impact of different alternative realisations of the P-ET model on discharge predictions, and to show how these alternative realisations*

*differ from the one realisation that correspond to true observations. The RR model used should be considered as a tool to demonstrate this, it is not intended to make predictions (although, studies for the Flemish Environmental Agency have shown that the statistics of the discharge series derived using the Uccle time series are similar to those of forcings measured near the catchment, and to those calculated on the actual discharge time series).*

Moreover, 1 year of data for evaluation is not enough. You will need a couple of years to ensure calibrated model can capture different aspects of a catchment.

*We understand the comment, though we believe it is not relevant. The PDM is in this study used as a tool to convert forcings into discharge, and the same model is used for both observed forcings (in Uccle) as modelled forcings. Yet, to make it realistic, we made use of a calibrated model, which indeed is not fully assessed for all possible hydrological circumstances. However, this model has shown to be able to well mimick the discharge behaviour of the Grote Nete catchment (as well as many other catchments in Flanders) and is used in operational water management. We believe that the model sufficiently mimicks the different hydrologic processes to show the potential of the developed generator for water management purposes. We will add some text explaining that the model used is sufficient for answering the objectives of the paper.*

5. Section 2.3: Copulas characterize dependence structure of different variables. This means there should be a dependence structure. Did you quantify the correlation between evaporation, temperature, and precip? If so, is it significant? At what temporal scale? My understanding is that you perform your analysis at daily scale, and I fear the correlation might not be significant at the daily scale.

   *We do not fully grasp the fear of the reviewer, but the correlation between the variables has been checked. Yet, the dependence structures, as is apparent in the input time series, are used to build the copula models.*

6. Line 150: bivariate → It could be multivariate

   *True, equation (1) can be written with more than two dimensions. However, as equation (1) only concerns 2 variables, we maintain the used terminology (i.e. bivariate copula).*

7. Line 152: I would reference to Joe 1997 too. Joe and Nelsen both played an important role in introducing copula to the scientific community.

   *We will add a reference to Joe (1997).*

8. Lines 171-173: I agree that vine copulas are very flexible, but it comes at a price! A model with 4 degrees of freedom is more flexible than a competitor with 2! However, usually there is not enough information to constrain all parameters. The copula literature usually does not address the parameter uncertainties, and so they neglect the identifiability of parameters. I would address this predicament here. For more info, refer to Figure 6 of: Sadegh, M., E. Ragno, and A. AghaKouchak (2017), Multivariate Copula Analysis Toolbox (MvCAT): Describing dependence and underlying uncertainty using a Bayesian framework, Water Resour. Res., 53, doi:10.1002/2016WR020242.
   Link: http://onlinelibrary.wiley.com/doi/10.1002/2016WR020242/full

   *We include this remark in the paper and make a reference to the paper suggested.*

9. Line 191: As a minor issue, when someone talks about a 3-dimensional model, I expect the model to have three parameters. When someone talk about trivariate model, I expect a multivariate model that associates three variables.

   *In literature, the terminology n-dimensional copulas is commonly used for a copula that involves n variates. One wouldn't call the Frank copula a one-dimensional copula as it only has one parameter. We believe this is a misconception of the reviewer. We therefore prefer not to change the terminology.*

10. Line 203: How did you construct the marginal distribution? Empirical? Fitted distribution?

    *We used the empirical distributions to construct the marginal distributions. We will add this information in the manuscript.*

11. Eq. 3: how did you calculate inverse of the vine copula? Analytical or numerical?

    *The inversions, necessary for sampling a value out of the vine copula, were performed numerically.*

12. Line 253: pvalue larger than 0.05 or smaller?!

*The obtained p-values were larger than 0.05. More details about the theory and the p-values of the White test are given in Shepsmeier, 2015.*

13. Section 2.4: How did you calibrate the modified BartlettLewis (MBL) model, given the stochastic nature of precipitation prediction models? With stochastic models, usually summary statistics of data and simulation are compared, rather than original time series. For this purpose, approximate Bayesian computation is a great framework.

   *We will briefly mention in the paper how the model was calibrated. Actually, the model was taken from a previous study (Pham et al., 2013).*

14. Lines 338-344: I cannot disagree more! Forcing and model uncertainties are intertwined, and interact in a nonlinear manner. It is not as simple as you explained. You cannot simply use a RR model calibrated for one watershed to simulate runoff at another watershed! Tens (Hundreds) of papers are available on the regionalization topic, not many of them really provided a sound ground for transferring model parameters from one watershed to another! Worse is that authors assume this modeling uncertainty does not interact with the forcing uncertainty.

   *The comment of the reviewer is based on the misconception that the RR model is calibrated for one catchment and applied to another one. As stated before, Uccle is a city (not a catchment) where the headquarter of the Royal Meteorological Office of Belgium is located. At this place, the meteorological data are obtained. These data, which are statistically similar to those observed in the catchment of the Grote Nete (situated less then 100 km from Uccle), are then used in the model of the Grote Nete catchment to model its discharge. The advantage of the data at Uccle is their length (72 years). Such long time series near the Grote Nete are not available. Furthermore, the simulations are made for one single catchment (i.e. that of the Grote Nete), making use of a model calibrated for it. In this sense, the exercise done in the paper allows for partly reducing the uncertainty due to the use of PDM.*

15. Figure 22: 22 figures? Is that many figures really necessary when most of them don't provide any new info?

   *We are aware of the large number of figures, however, we believe that they all are necessary to fully get the point we wish to make. Yet, by showing only the results for some months instead of for all months of*

*the year, we can reduce the number of subplots per figure while keeping the same message. By doing this exercise, it may become possible to merge figures attributed each time to one case into one figure showing results for the three cases. "*

*We are aware of the large number of figures, however, we believe that they all are necessary to fully get the point we wish to make (unless we refer to results that are not shown, but this is quite annoying for readers trying to dig into the results). Of course, instead of showing all months of the year, we can reduce this to a couple of months.*

16. Lines 476-478: I have a hard time accepting this claim. If you generate a much longer synthetic (stochastic) forcing, then lets say predictions at a 100 years return period level improves. I accept this. But I cannot accept the general comment that longer fording data reduces overall uncertainties. What if I had to estimate a 500 years return period flow?

*The uncertainties that are reduced are the result of working with time series of a given length. This should be better framed in the conclusion: not all uncertainties reduce, only the ones that are due the limited length of the time series reduce, while those caused by the stochastic nature of the climatological variables remain. To answer to the question on what to do when a 500-year return period of discharge would be needed, then the advice should be to model discharge with forcing time series that are a multitude of the return period one aims for. In this case, one should model series of 5000 years or more.*

---

## Author Response (AR1)

We would first like to thank both reviewers for their profound comments and suggestions. In this rebuttal, we list the comments and suggestions raised by the reviewers and explain how we handled them in the revised manuscript. Our answers are listed in italic. Changes to the manuscript are given in boldface.

**1 Specific comments of the first reviewer**

**1.1 Major issues**

**1.1.1 Seasonal effects**

A copula models the dependence between two random variables $X_1$, $X_2$ with marginal distributions $F_1$ and $F_2$. Its parameters can be estimated from observations of these variables, $\boldsymbol{X}_t = (X_{1,t}, X_{2,t})$, $t = 1, \ldots, T$. The usual assumption for the validity of the estimate is that the data $\boldsymbol{X}_t$ are independent and identically distributed (iid). In particular, the distribution of $\boldsymbol{X}_t$ should not change with $t$, which is usually violated by climatic variables. The authors acknowledge that by fitting multiple models, one for each month. I am afraid this may not be enough, because climatic trends also exist within months. For example: in central Europe, the end of April is on average much warmer than the beginning of April. Suppose that additionally, the average precipitation is decreasing during April. Then high temperatures will likely coincide with low levels of precipitation and vice versa. A copula fitted to this data will show negative dependence, which merely reflects the two deterministic within-month trends working in opposite directions, but not the stochastic dependence between the time series. Whether or not within-month trends exist can be easily checked visually or by formal statistical tests (e.g., Harris and Sollis, 2003, Chapter 3). If they do exist, they should be accounted for on a finer time scale. Splitting the data into weeks or even days could be a solution, but significantly decreases the number of observations available for fitting the copula model. A good alternative is to center and scale the time series by their seasonal mean and standard deviation. More specifically, if $X_{j,d,y}$ denotes variable $j$ observed at day $d$ of year $y$, set

$$\tilde{X}_{j,d,y} = (X_{j,d,y} - \mu_{j,d})/\sigma_{j,d}, \tag{1}$$

where $\mu_{j,d}$ and $\sigma_{j,d}$ are the mean and standard deviation of $X_{j,d,y}$ , $y = 1, \ldots, 72$. If necessary, trends in the skewness of $\tilde{X}_{j,d,y}$ can be removed similarly using a Box-Cox transformation. This transformation is usually

sufficient to account for deterministic seasonal effects. We can now build a copula for the stochastic dependence in $\tilde{X}_t = \tilde{X}_{d+365(y1)}$ . Simulated data from this model can be transformed to the original scale by inverting (1).

*We checked the data for the existence of a within-month trend, and found that trends exists for temperature and evapotranspiration data. As advised by the referee, we standardized the data. We added a paragraph (in Section 2.1) in the revised manuscript w.r.t. standardization of the data.:*

**"In order to use the above-described data to fit copulas, the data should be independent and identically distributed (*iid*), indicating that the distribution of the data should not change with time. To this end, the time series is split into monthly series to which a vine copula model can be fitted. Hence, for each month a different model will be obtained. However, the data distributions can also change within the monthly series, i.e. a within-month trend may exist. Therefore, the daily distributions, each containing 72 observations were compared w.r.t. their equality within each month by means of an ANOVA test, when distributions were homoscedastic, a Welch ANOVA test (Welch, 1951) when distributions were heteroscedastic, or a Kruskal Wallis test (Kruskal and Wallis, 1952) when distributions were not-normal and heteroscedastic, at a significance level of 0.001. The results of these tests indicate that within-month trends exist for temperature and evapotranspiration, whereas no trend was found for precipitation. In order to meet the requirements of the data to be *iid*, temperature and evapotranspiration data were standardized as follows:**

$$x_{s,d,y} = \frac{(x_{d,y} - \mu_d)}{\sigma_d}, \tag{2}$$

**with $x_{s,d,y}$ the standardized value of temperature or evapotranspiration at day $d$ of year $y$, $x_{d,y}$ the original measured value of temperature or evapotranspiration at day $d$ of year $y$, $\mu_d$ and $\sigma_d$ the mean value and standard deviation of $x$ at day $d$."**

**1.1.2   Inter-serial dependence**

Even when the distribution of $\tilde{X}_t$ is the same for each $t$, subsequent observations of the time series may not be independent. Such data is called stationary which is less restrictive than *iid*. Typically, stationarity is sufficient to allow for valid estimation of the marginal distributions and copula of

$\tilde{X}_t$. But inference tools (like confidence intervals and goodness-of-fit tests) derived under the *iid* assumption are no longer valid. Another potential issue is that inter-serial dynamics can play an important role in applications. If so, these dynamics should be modeled explicitly. In the context of hydrological discharges, this is likely the case. Large discharges often occur when extreme weather conditions have been persistent for several days, and persistence is a sign of inter-serial dependence. A simple way to check whether such dependence is present is to look at the autocorrelation of the time-series, i.e., the correlation between $\tilde{X}_t$ and $\tilde{X}_{t1}$ (and their squares). If the correlation is small, one can test statistically whether it is zero. If there is dependence, there are two popular ways to capture it:

1. **Copula models**: This route is taken by the authors in 2.3.3, but only for the temperature variable. Similar models for the inter-serial dependence in evapotranspiration and precipitation should be employed in addition. If $F_{j,t,t1}$ is the joint distribution of $\tilde{X}_{j,t}$ and $\tilde{X}_{j,t1}$, the between-variables dependence can be modeled by a copula for the variables

$$U_{1,t} = F_{1,t|t1}(\tilde{X}_{1,t}|\tilde{X}_{1,t1}), \quad U_{2,t} = F_{2,t|t1}(\tilde{X}_{2,t}|\tilde{X}_{2,t1}), \quad (3)$$

where $F_{j,t|t1}$ is the conditional distribution of $\tilde{X}_{j,t}$ given $\tilde{X}_{j,t1}$.

2. **Classical time series models**: Classical time series models (see, e.g., Shumway et al., 2000, Chapter 3) assume that the variable $\tilde{X}_{j,t}$ is a linear combination of the preceding values ($t' < t$) and *iid* noise. For example, the autoregressive model of order $p$ is

$$\tilde{X}_{j,t} = \sum_{k=1}^{p} \phi_{j,k}\tilde{X}_{j,tk} + \epsilon_{j,t}, \quad (4)$$

where $\phi_{j,k}$ are model parameters and $\epsilon_{j,t}$ is *iid* noise with mean zero. The sequence $\epsilon_{j,t}$ is commonly called innovation or residual series. The stochastic between-variable dependence can then be captured by a copula model for $(\epsilon_{1,t}, \epsilon_{2,t})$ More complex models are required when $\tilde{X}_{j,t}$ is autocorrelated (Harris and Sollis, 2003, Chapter 8).

*We found an autocorrelation for temperature and evapotranspiration on the basis of a Ljung-Box Q-test. We took the autocorrelation into account by extending the vine $V_{TPE}$ with $E_{t-1}$, i.e. including the evatranspiration of the previous time step (see Section 2.3.2 of the revised manuscript):*

**"In order to avoid monthly effects, the temperature and evapotranspiration data were first standardized and a different C-vine copula model is used for each month. However, subsequent observations of the time series may not be independent, meaning that values within the time series may be autocorrelated. This is accounted for by extending the vine copula $V_{TPE}$ as used in Pham et al. (2016) with the evapotranspiration of the previous day ($E_p$). In this way a four-dimensional C-vine copula $V_{TPE_pE}$ is constructed for each month. "**

**1.1.3 Assessing the quality of the vine copula model**

There are multiple issues with how the quality of the model is evaluated:

1. To check the models validity, the authors merely look at the density/cdf of the observed and simulated values of a single time series. This is only weakly related to the vine copula model and not a good indicator for its fit. Under this measure, just simulating from the distribution $F_E$ (thereby assuming that $E$ is independent of $T$ and $P$ ) would lead to results that are at least as good as the ones from the vine copula. To adequately assess the quality of the dependence model, pair-wise comparisons should be made. For example, one can look at the scatterplots of observed and simulated pairs $(X_{1,t}, X_{2,t})$. Another alternative are contour plots of the estimated joint density of observed vs. simulated pairs. Such comparisons should be made for all variable combinations. Similarly, multivariate return periods should be considered instead of single-variable return periods (see, Salvadori et al., 2011).

   *We examined the values of Kendall's tau to check whether or not the dependence between the generated variables is maintained. By examining scatter plots between observed and simulated pairs, as the referee suggests, one would rather investigate whether the simulated values approximate the observed values, as in a prediction model. In this study, however, it is not the aim to match the observed values for each time step t as closely as possible, but rather mimic the behaviour of the observed time series w.r.t. statistics and extremes. We therefore did not include scatterplots in the revised manuscript.*

2. Figures 6 and 9 use empirical cumulative distribution functions (ECDF) instead of densities for no obvious reason. I advise against using

ECDFs because they suggest a misleading sense of closeness between distributions. Since ECDFs are necessarily monotone functions with boundary values 0 and 1, their shape is quite restricted. For example, the left panels of Figure 9(d) show that the distributions are different, but the ECDFS still look somewhat similar. But the corresponding densities would show almost no overlap and more clearly communicate the dissimilarity.

*We changed the figures and used densities instead of the ECDFs.*

3. In Section 4, the uncertainty in the simulation model is assessed for various degrees of data availability. From the spread of estimated densities in Figures 11-15, the authors conclude that uncertainty increases when a variable is not observed and needs to be simulated. This is likely true, but can not be inferred from these figures. The density plots for cases 1-3 are based on a different number of observations. The spread seen on 125000 simulations will naturally be larger than the spread on 50 observations even when the actual distribution is the same. Hence, the spreads should only be compared when they are based on the same number of simulations.

*In the revised manuscript, we kept our original approach in cases 1-3, as in this way we also take the stochasticity of the MBL and the temperature model into account. However, we also compared the extremes obtained for the different cases when one would use the same number of simulations, i.e. for each simulated time series of temperature, only one corresponding time series of evapotranspiration is generated in case 2, and for each simulated time series of precipitation, only one corresponding time series of temperature and one of evapotranspiration are generated. The spread obtained for this approach has been added to Figure 15 in the revised manuscript, and some explanatory text has been included in the manuscript:*

**" However, it should also be noted that the results for cases 2 and 3 are obtained on the basis of a wider ensemble of time series as compared to case 1 (2500 for case 2, and 125000 for case 3). In order to also compare the variations obtained on the basis of an equal number of time series within the ensemble (i.e. 50 time series), for each time series of observed (case 2) or simulated (case 3) precipitation, one corresponding time series of temperature and one corresponding time**

series of evapotranspiration are generated. The bottom panel of Fig.15 illustrates the extremes obtained on the basis of this ensemble of 50 time series of discharge. These results also show that most of the variation obtained in case 3 is due to the stochastic generation of precipitation."

**1.2 Minor issues**

1. Since vine copulas are the essential ingredient in your model, I suggest to indicate this in the abstract.

   *This has been indicated in the abstract:*

   **"In this paper, stochastically generated rainfall and coinciding evapotranspiration time series, generated by means of vine copulas, are used to force a simple conceptual hydrological model."**

2. p. 4, p. 165: If unconditional bivariate copulas are used (as is common), a vine copula is not a decomposition, but a construction. A decomposition is called non-simplified vine copula and involves conditional bivariate copulas (see, e.g., Stöber et al., 2013). I suggest to rephrase this sentence.

   *We rephrased this sentence:*

   **A flexible construction method for high-dimensional copulas, known as the vine copula construction, has been introduced in the work of Bedford and Cooke (2001, 2002), in which multivariate copulas, and hence the multivariate densities, are constructed as a product of bivariate copula densities.**

3. p. 4, l. 167: I suggest to change all types of dependence to a wide range of dependence structures. All types can only be modeled by a non-simplified vine copula.

   *This has been changed in the manuscript.*

4. p. 5, l. 179: What do you mean by C-vine copulas are easier to construct than D-vine copulas? In fact, any three-dimensional vine is both a C- and D-vine, which can be easily verified by re-arranging the vertices of the vine graph

   *This sentence has been removed from the manuscript.*

5. p. 6, l. 203 ff.: I am afraid a reader without prior knowledge of vine copulas will not understand your paragraph on how the model is estimated. Instead of your explanation, it should suffice to refer the reader to Aas et al. (2009).

*We believe that the reader should get some insight in how the vine is constructed and therefore, we revised this paragraph and added a reference to Aas et al. (2009):*

**"The construction of $V_{TPE_pE}$ is given as follows. First, values $(u_{T,j}, u_{P,j}, u_{E_p,j}, u_{E,j})$ of $U_T$, $U_P$, $U_{E_p}$ and $U_E$ are derived from the marginal distributions of respectively $T$, $P$, $E_p$ and $E$ ($j = 1, ..., n$ and $n$ is the number of data points), and are used to select and fit the bivariate copulas $C_{TP}$, $C_{TE_p}$ and $C_{TE}$. These bivariate copulas are conditioned on $U_T$ through partial differentiation as given in Eq. (5), resulting in the conditional cumulative distribution functions $F_{P|T}$, $F_{E_p|T}$ and $F_{E|T}$ .**

$$F_{P|T}(u_P|u_T) = \frac{\partial}{\partial u_T} \mathbf{C}_{TP}(u_T, u_P)$$

$$F_{E_p|T}(u_{E_p}|u_T) = \frac{\partial}{\partial u_T} \mathbf{C}_{TE_p}(u_T, u_{E_p}) \tag{5}$$

$$F_{E|T}(u_E|u_T) = \frac{\partial}{\partial u_T} \mathbf{C}_{TE}(u_T, u_E) .$$

**Using these three conditional distributions, the conditional probabilities are calculated for all data points $(u_{T,j}, u_{P,j}, u_{E_p,j}, u_{E,j})$. To these conditional probabilities, which are also uniformly distributed on [0,1], two bivariate copulas $C_{PE_p|T}(F_{P|T}, F_{E_p|T})$ and $C_{PE|T}(F_{P|T}, F_{E|T})$ are fitted, of which the partial derivatives to $F_{P|T}$ can be computed to obtain $F_{E_p|TP}$ and $F_{E|TP}$. Again, using these two conditional distributions, a bivariate copula $C_{E_pE|TP}(F_{E_p|TP}, F_{E|TP})$ is fitted, which can also be conditioned by calculating the partial derivative. For more detailed information about the construction of vine copulas, we refer to (Aas et al., 2009)."**

6. How are marginal distributions modeled/estimated?

*We added in the manuscript that empirical distributions are employed as marginal distributions.*

7. Figures 4, 12–17 should use a larger smoothing parameter to decrease variability of the density estimates. A large proportion of the observed

variability is due to the density estimation technique. This is not the kind of variability you want to assess.

*We changed the smoothing parameter to reduce this variability.*

8. Figure 22: What is i?

*i is the cumulative relative frequency at which the RMSD between the simulated and the reference discharge is calculated. See also the explanation of Eq. (6):*

$$\text{RMSD}(i) = \sqrt{\frac{1}{n} \sum_{s=1}^{n} \left(Q_{m,s}(i) - Q_o(i)\right)^2}, \tag{6}$$

*where $Q_m(i)$ and $Q_o(i)$ are respectively the modelled and reference discharge value at a cumulative relative frequency $i \in [0, 1]$, and $n$ is the number of the members in the ensemble considered.*

**2  Comments of the second reviewer**

This is a nice study that uses copula-based approaches to stochastically generate mutually dependent rainfall and evapotranspiration forcing for rainfall runoff models. This could be used for design purposes where observation is sparse or missing. However, this approach does not seem to be working properly for the extreme events (which are needed for design purposes). Acknowledging this fact, the study is valuable for the areas with no observation. Overall, paper is well written and well structured. I have some comments (most of them major) that could potentially improve the quality of this paper.

1. Line 46: Authors use stochastic process models to generate precipitation series. My question is:
   How do stochastic process models handle changing characteristics of precipitation? Several studies have shown, for parts of the world, that rainfall events are shrinking in time and expanding in amplitude. Also there is a temporal shift in rainfall events in some parts of the world, let alone the changes in the distribution of rainfall/snow. Addressing these issues could be helpful.

   *We addressed this in the introduction:*

**"The BL model will be employed on a monthly basis such that temporal changes in precipitation characteristics due to the annual cycle can be underpinned. Long-term changes, e.g. due to climate change, however, cannot be accounted for in this model set-up."**

2. Lines 91-94: I don't understand how the number of stochastically generated forcing data could influence the uncertainty of the rainfall-runoff model's response. Uncertainty is a characteristic of the forcing data (let's neglect the modeling uncertainties for now), not the number of generated time series. So if you find a time series that fit your runoff extremes well, this is just a random phenomenon. This cannot be the basis for prediction, as we can't determine the best forcing for future, and need to rely on the ensemble of forcing data.

   *We added some text in the introduction to describe the source of uncertainty we are dealing with:*

   **"By regarding the actual observed time series as one realisation of the meteorological process, the corresponding discharge can also be regarded as one realisation. Actually, due to chaos occurring in the climatological system, a different time series could have been observed resulting in a discharge time series different from the actual observed one. The latter will hence provide other design values than those corresponding to the actual observed time series. In order to account for this kind of uncertainty, different cases, in which the number of stochastically generated input variables to the model is increased, are investigated. For these cases, the increase of uncertainty in modelled extremes and what portion of this increase can be attributed to the different stochastic generators, is assessed."**

3. Lines 95-96: Section 2 should precede section 3!

   *This has been corrected.*

4. I am confused about how sections 2.1 and 2.2 are connected. Historical record of climate forcing are obtained for Brussels, and RR model is calibrated for the Grote Nete catchment. How do you use a model calibrated against one watershed, to predict runoff at another watershed? Moreover, 1 year of data for evaluation is not enough. You will

need a couple of years to ensure calibrated model can capture different aspects of a catchment.

*We added some text in Section 2.2. to better explain this.:*

**"Given the relatively small distance between Uccle and the Grote Nete catchment, and the fact that the meteorological conditions are nearly the same, one can assume that the statistics of the modelled discharge obtained with the forcing data observed near the catchment and those observed at Uccle are negligible. Furthermore, the rainfall-runoff model will not be used to make predictions, but rather to demonstrate the impact of different alternative realisations of precipitation ($P$), temperature ($T$) and evapotranspiration ($E$) on discharge values. Therefore, although PDM will be applied to observations from Uccle in this study, it is calibrated on the basis of a time series of more than 6 years (from 13/8/2002–31/12/2008) at an hourly time-step (precipitation, evapotranspiration and discharge) that is available for the catchment."**

5. Section 2.3: Copulas characterize dependence structure of different variables. This means there should be a dependence structure. Did you quantify the correlation between evaporation, temperature, and precip? If so, is it significant? At what temporal scale? My understanding is that you perform your analysis at daily scale, and I fear the correlation might not be significant at the daily scale.

   *We do not fully grasp the fear of the reviewer, but the correlation between the variables has been checked. Yet, the dependence structures, as present in the input time series, are used to build the copula models.*

6. Line 150: bivariate → It could be multivariate

   *True, Eq. (1) can be written with more than two dimensions. However, as Eq. (1) only concerns 2 variables, we maintain the used terminology (i.e. bivariate copula).*

7. Line 152: I would reference to Joe 1997 too. Joe and Nelsen both played an important role in introducing copula to the scientific community.

   *We added a reference to Joe (1997).*

8. Lines 171-173: I agree that vine copulas are very flexible, but it comes at a price! A model with 4 degrees of freedom is more flexible than a competitor with 2! However, usually there is not enough information to constrain all parameters. The copula literature usually does not address the parameter uncertainties, and so they neglect the identifiability of parameters. I would address this predicament here. For more info, refer to Figure 6 of: Sadegh, M., E. Ragno, and A. AghaKouchak (2017), Multivariate Copula Analysis Toolbox (MvCAT): Describing dependence and underlying uncertainty using a Bayesian framework, Water Resour. Res., 53, doi:10.1002/2016WR020242.
Link: http://onlinelibrary.wiley.com/doi/10.1002/2016WR020242/full

*We added a sentence to make the reader aware of this:*

**" However, one has to be aware that the flexibility offered by vine copulas demands the estimation of a large number of parameters for which the data set should encompass sufficient information."**

9. Line 191: As a minor issue, when someone talks about a 3-dimensional model, I expect the model to have three parameters. When someone talk about trivariate model, I expect a multivariate model that associates three variables.

*We prefer not to change the terminology as, in literature, the term 'n-dimensional copula' is commonly used for a copula that involves n variables. One wouldn't call the Frank copula a one-dimensional copula as it only has one parameter.*

10. Line 203: How did you construct the marginal distribution? Empirical? Fitted distribution?

*We used the empirical distributions to construct the marginal distributions. We added this information in the manuscript.*

11. Eq. 3: how did you calculate inverse of the vine copula? Analytical or numerical?

*The inversions, necessary for sampling a value out of the vine copula, were performed analytically whenever possible, numerically otherwise.*

12. Line 253: pvalue larger than 0.05 or smaller?!

*The obtained p-values were larger than 0.05. More details about the theory and the p-values of the White test are given in Shepsmeier,*

*2015. We added some more information about the hypothesis of the White test in the paper:*

**"Further, the White goodness-of-fit test (Schepsmeier, 2015) is applied to check whether the dependence present in the data is captured by the C-vine copulas. For this test, $p$-values larger than the significance level indicate that the dependence structure of the data can be described by the selected copulas."**

13. Section 2.4: How did you calibrate the modified BartlettLewis (MBL) model, given the stochastic nature of precipitation prediction models? With stochastic models, usually summary statistics of data and simulation are compared, rather than original time series. For this purpose, approximate Bayesian computation is a great framework.

    *We briefly mentioned in the paper how the model was calibrated:*

    **"The MBL model is calibrated using the Generalised Method of Moments, i.e. the difference between the model statistics obtained by means of analytical expressions and the empirical statistics obtained from the observed time series is to be minimized. The calibration of the MBL model in this study is based on the mean, variance, lag-1 autocovariance and zero-depth probability (ZDP) at the aggregation levels of 24 h, 48 h and 72 h instead of 10 min, 1 h and 24 h that were used in Pham et al. (2013). As in Pham et al. (2013), the Shuffled Complex Evolution algorithm Duan et al. (1994) was employed to search for the optimal parameters.**

14. Lines 338-344: I cannot disagree more! Forcing and model uncertainties are intertwined, and interact in a nonlinear manner. It is not as simple as you explained. You cannot simply use a RR model calibrated for one watershed to simulate runoff at another watershed! Tens (Hundreds) of papers are available on the regionalization topic, not many of them really provided a sound ground for transferring model parameters from one watershed to another! Worse is that authors assume this modeling uncertainty does not interact with the forcing uncertainty.

    *The comment of the reviewer is based on the misconception that the RR model is calibrated for one catchment and applied to another one. As stated before, Uccle is a city (not a catchment) where the headquarter of the Royal Meteorological Office of Belgium is located. At this place, the*

*meteorological data are obtained. These data, which are statistically similar to those observed in the catchment of the Grote Nete (situated less then 100 km from Uccle), are then used in the model of the Grote Nete catchment to model its discharge. The advantage of the data at Uccle is their length (72 years). Such long time series near the Grote Nete are not available. Furthermore, the simulations are made for one single catchment (i.e. that of the Grote Nete), making use of a model calibrated for it. In this sense, the exercise done in the paper allows for partly reducing the uncertainty due to the use of PDM.*

15. Figure 22: 22 figures? Is that many figures really necessary when most of them don't provide any new info?

    *We reduced the number of figures. Figures 12–18 of the original manuscript have now been summarized in three figures.*

16. Lines 476-478: I have a hard time accepting this claim. If you generate a much longer synthetic (stochastic) forcing, then lets say predictions at a 100 years return period level improves. I accept this. But I cannot accept the general comment that longer fording data reduces overall uncertainties. What if I had to estimate a 500 years return period flow?

    *The uncertainties that are reduced are the result of working with time series of a given length. We rephrased this sentence in the conclusion:*

    **"With respect to extreme discharge, it was shown that the uncertainties encountered in case 3 are partly caused by the limited length of the time series used. The uncertainties on the predictions highly reduce when input time series are used that are much longer than the maximum return period aimed at. As in this particular case, all forcing data are generated, the modeller is not restricted to the length of an observed time series, and can hence generate time series of whatever length as input to the hydrological model, taking into account that the longer the time series used, the more the uncertainty reduces at the expense of increasing run-time."**

    *To answer to the question as what to do when a 500-year return period of discharge would be needed, then the advice should be to model discharge with forcing time series that are a multitude of the return period one aims for. In this case, one should model series of 5000 years or more.*

**References**

Aas, K., Czado, C., Frigessi, A., and Bakken, H. (2009). Pair-copula constructions of multiple dependence. *Insurance: Mathematics and Economics*, 44(2):182–198.

Bedford, T. and Cooke, R. M. (2001). *Monte Carlo Simulation of Vine Dependent Random Variables for Applications in Uncertainty Analysis.* Management Science, theory, method and practice series. University of Strathclyde, Department of Management Science.

Bedford, T. and Cooke, R. M. (2002). Vines–a new graphical model for dependent random variables. *The Annals of Statistics*, 30(4):1031–1068.

Duan, Q., Sorooshian, S., and Gupta, V. K. (1994). Optimal use of the SCE-UA global optimization method for calibrating watershed models. *Journal of Hydrology*, 158(3–4):265–284.

Kruskal, W. H. and Wallis, W. A. (1952). Use of ranks in one-criterion analysis of variance. *Journal of the American Statistical Assocciation*, 47:583–621.

Pham, M. T., Vanhaute, W. J., Vandenberghe, S., De Baets, B., and Verhoest, N. E. C. (2013). An assessment of the ability of Bartlett–Lewis type of rainfall models to reproduce drought statistics. *Hydrology and Earth System Sciences*, 17(12):5167–5183.

Pham, M. T., Vernieuwe, H., De Baets, B., Willems, P., and Verhoest, N. E. C. (2016). Stochastic simulation of precipitation-consistent daily reference evapotranspiration using vine copulas. *Stochastic Environmental Research and Risk Assessment*, 30(8):2197–2214.

Schepsmeier, U. (2015). Efficient information based goodness-of-fit tests for vine copula models with fixed margins: A comprehensive review. *Journal of Multivariate Analysis*, 138:34–52.

Welch, B. (1951). On the comparison of several mean values: an alternative approach. *Biometrika*, 38(3–4):330–336.